# Neuroanatomical abnormalities in a nonhuman primate model of congenital Zika virus infection

Danielle Beckman[1†], Adele MH Seelke[1,2†], Jeffrey Bennett[1,2], Paige Dougherty[1,2], Koen KA Van Rompay[1,3], Rebekah Keesler[1], Patricia A Pesavento[3], Lark LA Coffey[3], John H Morrison[1,4], Eliza Bliss-Moreau[1,2]*

[1]California National Primate Research Center, UC Davis, Davis, United States; [2]Department of Psychology, UC Davis, Davis, United States; [3]Department of Pathology, Microbiology and Immunology, School of Veterinary Medicine, UC Davis, Davis, United States; [4]Department of Neurology, School of Medicine, UC Davis, Davis, United States

**Abstract** We evaluated neuropathological consequences of fetal ZIKV exposure in rhesus monkeys, a translatable animal model for human neural development, by carrying out quantitative neuroanatomical analyses of the nearly full-term brains of fetuses infected with ZIKV and procedure-matched controls. For each animal, a complete cerebral hemisphere was evaluated using immunohistochemical (IHC) and neuroanatomical techniques to detect virus, identify affected cell types, and evaluate gross neuroanatomical abnormalities. IHC staining revealed the presence of ZIKV in the frontal lobe, which contained activated microglia and showed increased apoptosis of immature neurons. ZIKV-infected animals exhibited macrostructural changes within the visual pathway. Regional differences tracked with the developmental timing of the brain, suggesting inflammatory processes related to viral infiltration swept through the cortex, followed by a wave of cell death resulting in morphological changes. These findings may help explain why some infants born with normal sized heads during the ZIKV epidemic manifest developmental challenges as they age.

**\*For correspondence:**
eblissmoreau@ucdavis.edu

[†]These authors contributed equally to this work

**Competing interest:** The authors declare that no competing interests exist.

## Editor's evaluation

This rigorous study provides compelling evidence that Zika virus infections in infants can markedly impact brain development through neuroinflammatory mechanisms. The work will have broad interest among developmental neurobiologists, as well as scientists whose work focuses on ZIKV pathogenesis.

## Introduction

In late 2015, medical professionals in the northeast region of Brazil reported a surge in the number of children born with microcephaly, with an increase of 265% in the number of cases during 2015–2016 (*Microcephaly Epidemic Research Group, 2016*; *Kleber de Oliveira et al., 2016*). Infants born with microcephaly during the Zika outbreak also had a constellation of clinical presentations that ultimately became known as Congenital Zika Syndrome (CZS): skull deformation, abnormally small cerebral cortices, and retinal scarring, among others (*Broussard et al., 2018*; *Moore et al., 2017*; *Musso et al., 2019*). Clinicians have identified additional health issues as these children have continued to grow and develop (*Wheeler et al., 2018*; *Cardoso et al., 2019*), including visual impairment (*Ventura et al., 2017*), increased risk for autism (*Vianna et al., 2018*), atypical motor development, increased

risk for cerebral palsy (*Marques et al., 2019*), and poor sleep quality (*Pinato et al., 2018*). Further screening has identified cases where children not diagnosed with CZS at birth exhibit CZS by their first birthday; up to 50% of children born to mothers infected with Zika virus during pregnancy infection exhibit anatomical or behavioral abnormalities (*Cardoso et al., 2019*; *Wheeler, 2018*). Taken together, these findings suggest that while microcephaly is an extreme manifestation of CZS, children born with normal sized heads may also have other significant neural deficits. Understanding the anatomy of those deficits is critical for predicting the developmental trajectories of children with CZS and ultimately for developing effective treatments and interventions, and the focus of the present report.

Since the onset of the ZIKV pandemic that began in 2015, a number of studies have revealed important information about how ZIKV targets specific cell types and proteins within the central nervous system (for reviews *Khaiboullina et al., 2019*; *Sutarjono, 2019*). The neurotropism of the virus may result from ZIKV using AXL, a tyrosine kinase receptor important in modulating the innate immune system (*Rothlin et al., 2007*; *Hafizi and Dahlbäck, 2006*), to enter the brain (*Meertens et al., 2017*; *Rossi et al., 2020*; *Nowakowski et al., 2016*). Single-cell RNA sequencing analyses have demonstrated that AXL is highly expressed in radial glia, endothelial cells, microglia, and astrocytes in the developing human cortex (*Nowakowski et al., 2016*). After breaching the placental–fetal barrier, ZIKV infects radial glial cells. Radial glia cells undergo symmetric and asymmetric cell division to generate the neurons that populate the cerebral cortex (*Anthony et al., 2004*; *Falk and Götz, 2017*), and the disruption of this process may ultimately lead to decreased neuronal numbers resulting in microcephaly. Although ZIKV might not be cytotoxic for microglia and astrocytes initially, once infected, these cells can propagate and spread the virus through the brain, maintaining a high viral load in the brain over time (*Muffat et al., 2018*).

An increasing number of studies point to the involvement of glial cells in ZIKV infection of the CNS. Microglia are well known for their role as cellular effectors of innate immunity; together with astrocytes, they drive the neuroinflammatory process through phagocytosis and cytokine release. They also play a critical role in neural development by promoting neuronal survival via release of neurotrophic factors to support neuronal circuit formation, phagocytosing immature neurons that fail to form proper neuronal circuits, and removing redundant or dysfunctional synapses in the developing brain (*Thion et al., 2018*; *Reemst et al., 2016*). During the late stages of brain development and after birth, glial cells develop and spread through the nervous system, constituting at least half of the cellular population within the brain (*Zuchero and Barres, 2015*). Two possible pathways exist by which ZIKV may interact with glial cells –either direct infection or induction of an inflammatory response that activates microglia and/or astrocytes. Congenitally acquired ZIKV has been shown to infect glial precursor cells, impairing their distribution in the brain, leading to reduced and delayed myelination (*Chimelli and Avvad-Portari, 2018*), ultimately contributing to neuroinflammation and microcephaly.

Nonhuman primate (NHP) models of ZIKV infection have been established and are able to bridge the translational divide between rodent models and humans (*Nguyen et al., 2017*; *Coffey et al., 2018*). Rodent models have significant limitations including the fact that adult mouse models of ZIKV must be immunocompromised or genetically modified (e.g., *Winkler and Peterson, 2018*). Fetal and neonatal wild-type mice do show some susceptibility to ZIKV infection, but they must be directly exposed to the virus, either through intracerebroventricular, intraperitoneal, or intra-amniotic (IA) inoculation, and as such are not a good model for vertical transmission of ZIKV from infected mothers to fetuses (e.g., *Paul et al., 2018*). In addition, the most severe symptoms of CZS include dysfunction in the neocortex, especially the prefrontal cortex (PFC), and the extent to which certain cortical regions are homologous in rodents and primates is not clear (*Laubach et al., 2018*). NHPs gestational development is also similar to humans, including its extended duration (relative to rodents), the prevalence of singleton pregnancies (rather than litters), and the structure of the placenta (*Carter, 2007*). The anatomical development and organization of the brain also proceed along a similar path in humans and NHPs (*Workman et al., 2013*). Similar to humans, pregnant NHPs do not need to be immunocompromised to be infected with ZIKV, experience viremia, and demonstrate transplacental transmission of the virus to the fetus (*Koide et al., 2016*; *Dudley et al., 2016*; *Morrison and Diamond, 2017*; *Coffey et al., 2017*). Although none had evidence of microcephaly, previous studies of ZIKV-exposed fetal and infant macaques have identified a number of pathological features, including calcifications, abnormal gliosis, and white matter hypoplasia (*Coffey et al., 2018*; *Mavigner et al., 2018*; *Adams*

*Waldorf et al., 2018*; *Adams Waldorf et al., 2016*). Studies that employed immunohistochemical (IHC) analysis also identified loss of neural precursors, reduced neurogenesis, gliosis, and increased apoptosis in different brain regions (*Coffey et al., 2018*; *Mavigner et al., 2018*; *Adams Waldorf et al., 2018*). Neuroanatomical evaluations in these studies have adopted approaches that are standard for pathologic analyses, insofar as they typically evaluate small areas of brain tissue and use formalin fixation and paraffin embedding, procedures that can mask epitopes, reduce antigenicity, and make it challenging to understand the impact of infection on the whole brain (*Cinar et al., 2006*; *Jones and Hartman, 1978*).

The present study builds upon a previously described macaque model of CZS in which, to assure fetal infection at a defined time of gestation, fetuses were inoculated with ZIKV via the IA routes, mothers were concurrently inoculated with ZIKV intravenously (IV), and fetuses were harvested at the end of gestation (*Coffey et al., 2018*). We elected to use IA inoculation to ensure that the fetuses were infected so that we could study fetal outcomes, rather than inoculating the mother only and leaving open the possibility that fetuses did not become infected (i.e., at the time we initiated our study it was not clear whether vertical transmission rates were 100% although there is some subsequent evidence that is the case [*Nguyen et al., 2017*]). While RT-qPCR-based virology, immunology, and histology findings on these animals have been described previously (*Coffey et al., 2018*), the current report describes a more in-depth neuropathological assessment of the macro- and microstructural effects of ZIKV infection on the developing fetal brain using quantitative microscopy, comparing the effects of ZIKV infection in a developed cortical region (at the caudal extent of the brain, including the occipital lobe, Area 17) and an immature cortical region (at the rostral extent of the brain, including the frontal lobe, Area 46). Critically, we maintained the spatial integrity of the tissue and examined whole tissue sections across the complete hemisphere available for study; as such the analyses presented here differ from previous reports because we quantified specific anatomical regions of the brain (e.g., Areas 17 and 46) including features related to their macrostructure (e.g., cortical thickness) and microstructure (e.g., morphological analyses; identification of immature neurons in the process of apoptosis, etc.). These methods allowed us to identify where the virus was found, what cell types were affected, and whether gross neuroanatomical abnormalities were present.

## Results

To identify and characterize neuroanatomical consequences of fetal ZIKV, we examined the brains of six near full-term monkeys, including three animals that had been inoculated with ZIKV at different gestational days (GD 50, 64, and 90; full term is approximately GD 165) and three procedure-matched controls (*Coffey et al., 2018*). The pregnant animal inoculated at GD 64 spontaneously gave birth to a small but viable baby on GD 151, at which point both mother and baby were euthanized. The remaining ZIKV-infected subjects and matched controls were monitored until GD 155 when fetectomies were performed followed by necropsy (*Coffey et al., 2018*). The left cerebral hemisphere was used for both analysis of viral load and histology (*Coffey et al., 2018*) while the right hemisphere was preserved for anatomical analyses. The right hemisphere was sectioned into four blocks to allow for more efficient immersion fixation and cryoprotection (see *Figure 1A*). Brains were sectioned on a freezing sliding microtome, and one series underwent Nissl staining following standard lab protocols (*Kreutzberg, 1984*; *Lavenex et al., 2009*). Nissl-stained sections from the most anterior block and most posterior block were evaluated to determine if there were gross anatomical abnormalities and if development had proceed as expected (from caudal to rostral; *Workman et al., 2013*). The frontal and occipital lobes were selected as initial targets for evaluation because neocortex develops along the caudal–rostral axis, and at the day of birth the occipital lobe exhibits a well-defined pattern of cortical lamination while the layers in the PFC are indistinct (*Workman et al., 2013*; *Silbereis et al., 2016*). Additionally, we visually inspected each Nissl-stained tissue section to see if there were additional macrostructural abnormalities.

### Neuroinflammatory abnormalities and ZIKV presence in brains from ZIKV-infected animals

IHC confocal microscopy revealed the presence of clusters of reactive glia within Area 46 of the PFC from ZIKV+ animals, but not in the controls (*Figure 1A, B*). Multiple labeling fluorescent microscopy

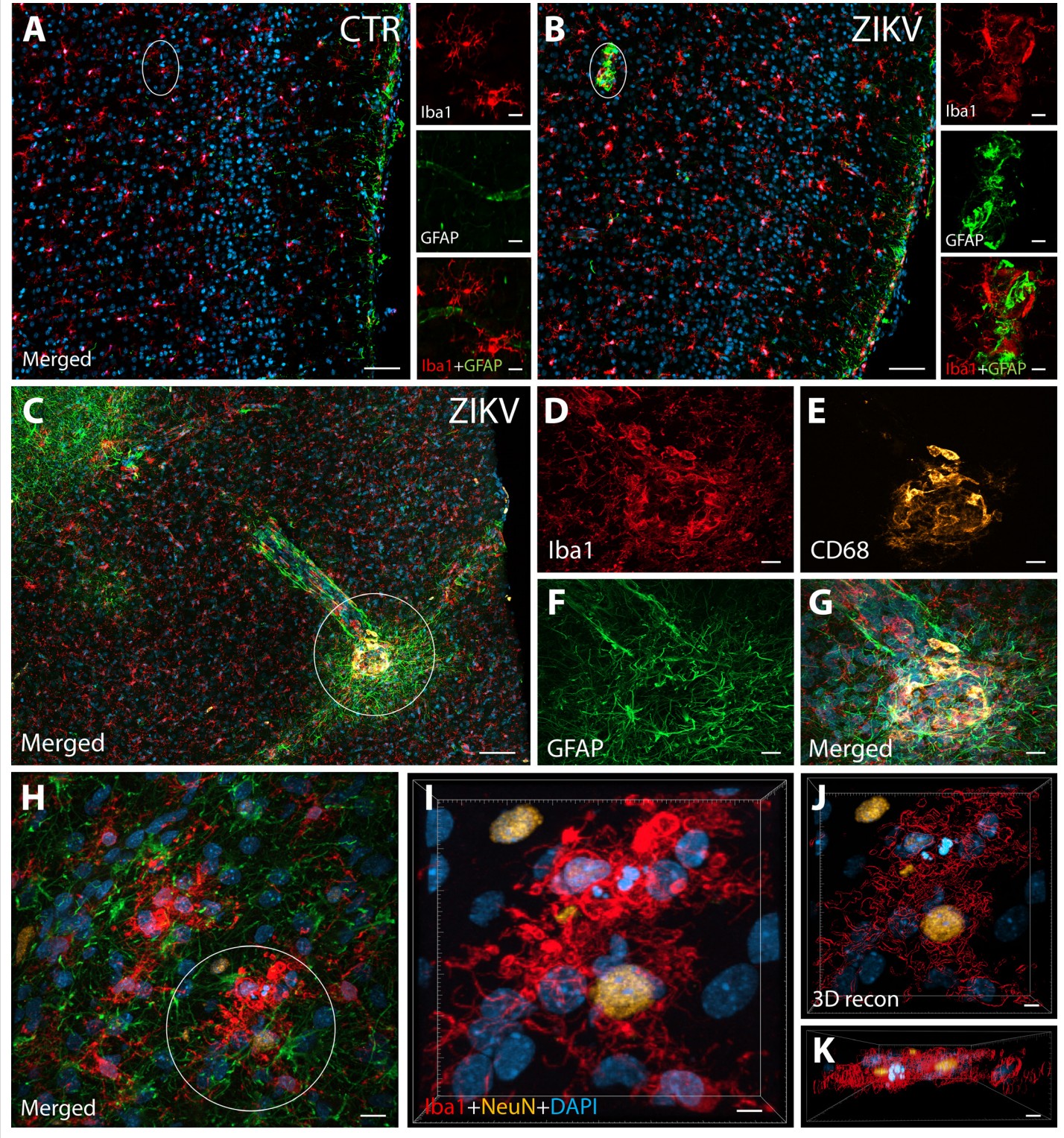

**Figure 1.** ZIKV infection induce neuroinflammatory response associated with activated microglia in immature prefrontal cortex. Unlike control animals (**A**), the tissue of the ZIKV-infected animals (**B**) evidenced clusters of round-shaped microglia (Iba1, red) and astrocytes ( Glial fibrillary acidic protein (GFAP), green) within selective cortical areas. Scale bar: 50 μM, zoom: 10 μM. In the Area 46 of the ZIKV-infected animals, the microglia marker Iba1 (red), colocalized with lysosomal protein CD68 (orange), a marker for phagocytic activity normally observed in activated microglia (**C–G**). Scale bar: 100 μM, zoom: 5 μM. Activated microglia surrounding apoptotic neurons (NeuN, orange) are shown within Area 46 in ZIKV animals (**H**, highlighted in **I**). 3D analysis of the micrographs shows apoptotic material (fragmented 4',6-diamidino-2-phenylindole (DAPI) and NeuN) inside activated microglia (**J**, **K**, scale bar: 5 μM).

allowed us to visualize that abnormal clusters of Iba1+ microglia which also express the lysosomal marker CD68 in the ZIKV-infected animals (*Figure 1C–G*). CD68 levels are known to be substantially upregulated in microglia during inflammatory processes and are thought to be involved in active phagocytosis (*Jurga et al., 2020*). We also observed that clusters of amoeboid-shaped microglia within Area 46, are directly engulfing neuronal debris in the ZIKV-infected monkeys (*Figure 1H, I*; 3D reconstruction: *Figure 1J, K*).

Next, we sought to detect the presence of ZIKV protein in the same regions we observed active inflammatory response. Interestingly, we observed the presence of ZIKV envelope protein within Area 46 of all the infected animals, with detection of viral protein inside clusters of microglia, but not astrocytes (GFAP, *Figures 2A–G* and 3D reconstruction: *Figure 2H*). Despite the fact that cells expressing ZIKV were detected in Area 46, we were not able to confirm the identity of the infected cells. As shown in *Figure 2I, J*, these ZIKV+ cells are thought to be in a direct process of phagocytosis, a common event occurring during viral encephalitis (*Chhatbar and Prinz, 2021*). While we could not detect the expression of other proteins in the ZIKV+ cells, they were surrounded and in direct contact with reactive microglia also expressing CD68, as shown in *Figure 2K, L*.

## ZIKV-induced changes to the occipital and frontal lobe

### Macrostructural changes

The most prominent manifestation of CZS is microcephaly (*Moore et al., 2017*), but none of the ZIKV-infected animals studied here exhibited a disproportionately (compared to their bodies) smaller biparietal diameter during gestation or head sizes at necropsy (*Coffey et al., 2018*). However, in less severe cases of CZS there have been documented changes to gross anatomical features of the brain including gyral simplification (defined as a reduction in the number of gyri associated with shallow sulci) and lissencephaly. Given those findings in humans, an analysis of the surface-area-to-volume ratio (or gyrencephality) of the occipital and frontal lobes from our infected and control animals allowed for an evaluation of these more subtle (compared to microcephaly) changes related to infection (*Figure 3*). These analyses revealed important differences in macrostructural features in Zika-infected versus control brains.

Gyrification of the occipital lobe was significantly reduced in ZIKV-infected animals as compared with control animals ($t_4$ = 3.33, p = 0.029; $d$ = 2.72) (*Figure 3B*). Within the occipital lobes there were no significant differences between ZIKV and control animals in the total lobe volumes ($t_4$ = 0.46, p = 0.67; $d$ = 0.37) or white and gray matter volumes ($t_4$ = 1.33, p = 0.25; $d$ = 1.09 and $t_4$ = 0.264, p = 0.805; $d$ = 0.22, respectively). However, differences between the proportions of white and gray matter varied by condition. The occipital lobes from ZIKV animals had a significantly higher proportion of white matter and lower proportion of gray matter than the control animals ($t_4$ = 2.93, p = 0.043; $d$ = 2.39; because the occipital lobe included only gray and white matter, the proportions are inverses of each other and the statistics identical) (*Figure 3C*). To evaluate cortical thickness, we selected one clearly defined anatomical region completely contained within the occipital lobe – Brodmann's Area 17. Cortical thickness in that area did not differ significantly between ZIKV and control subjects ($t_4$ = 1.42, p = 0.230; $d$ = 1.16; *Figure 3A*), although all the data points for the ZIKV group were below the mean of the control group, suggesting the possibility of cortical thinning within the primary visual cortex.

There was no difference between ZIKV and control animals in the gyrification of the frontal lobes ($t_4$ = 0.56, p = 0.60; $d$ = 0.46) (*Figure 3B*). The total volume of the frontal lobes did not differ between ZIKV and control animals ($t_4$ = 1.20, p = 0.30; $d$ = 0.98), and neither did the white and gray matter volumes ($t_4$ = 0.35, p = 0.75; $d$ = 0.28 and $t_4$ = 1.53, p = 0.20; $d$ = 1.25, respectively), nor relative proportions of white and gray matter ($t_4$ = 1.80, p = 0.15; $d$ = 1.47 and $t_4$ = 0.59, p = 0.59; $d$ = 0.48, respectively; *Figure 3D*). As in the occipital lobe analyses, we selected one anatomical region completely contained within the frontal lobe – Brodmann's Area 46 – and carried out a cortical thickness evaluation of that entire area. There were no significant differences between cortical thickness in Area 17 (*Figure 4A*) and in Area 46 (*Figure 4B*) of ZIKV and control animals ($t_4$ = 0.57, p = 0.60; $d$ = 0.46; *Figure 4B*). Taken together, these findings suggest that ZIKV infection did not cause gross morphological changes to frontal cortex during development.

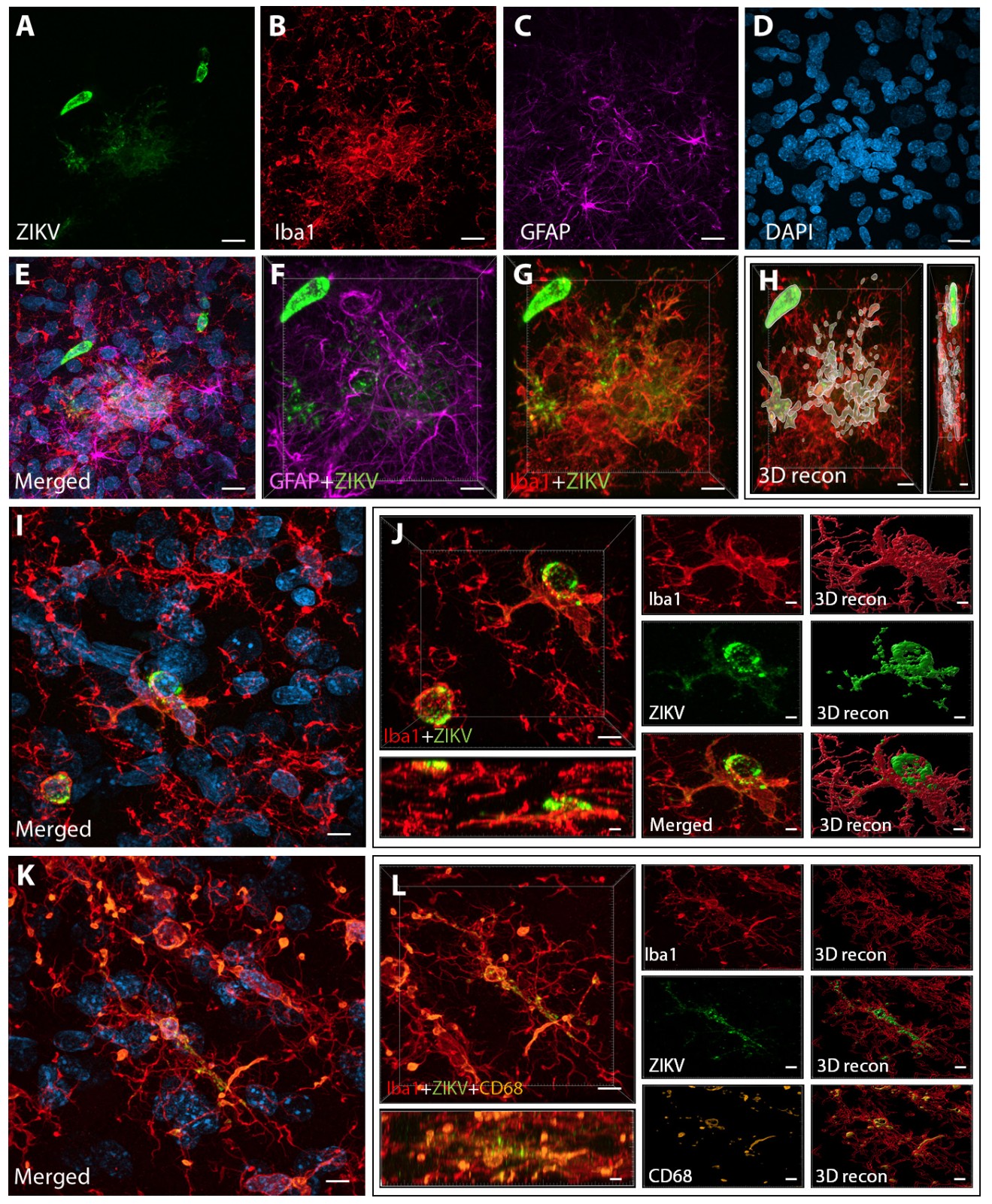

**Figure 2.** Intra-amniotic inoculation of ZIKV results in active brain infections of fetuses up to 105-day postinoculation. ZIKV protein (antiflavivirus, green) (**A**) is identified within Area 46 of ZIKV inoculated animals. ZIKV+ cells are surrounded by activated microglia (Iba1, red) (**B**) and astrocytes (GFAP, purple) (**C**), with DAPI (**D**) staining all nuclei. Scale bar: 10 µM. 3D analysis and reconstruction of confocal micrographs obtained in Area 46, highlight the presence of viral protein within the microglia (**E–H**). Scale bar: 10 µM, 3D: 5 µM. ZIKV+ cells observed in the same brain region are surrounded and

*Figure 2 continued on next page*

actively contacted by reactive microglia (**I, J**). Scale bar: 25 µM, zoom: 3 µM. Multiple labeling combining both microglia markers and viral protein shows the presence of viral protein inside phagocytic microglia in the prefrontal cortex (PFC) of the ZIKV-infected fetuses (**K, L**). Scale bar: 25 µM, zoom: 3 µM.

## M1 cortical thickness, and volumes of the amygdala, hippocampus, and optic tract

We carried out a few additional gross level analyses of one additional cortical target and three subcortical targets. We evaluated the cortical thickness of one additional cortical area, Area 4, primary motor cortex, was included because it is roughly in the middle of the brain (on the rostral–caudal axis) and thus represents a cortex at a different point in development compared to Areas 17 and 46. Three thickness measurements were taken on each of three different slides and then the value were averaged. There was not a difference in cortical thickness between the Zika and control animals in Area 4 ($t_4$ = 0.80, p = 0.47; $d$ = 0.65; $Mean_{ZIKV}$ = 2427.76 µm, $SD_{ZIKV}$ = 95.64 µm; $Mean_{Control}$ = 2361.72 µm, $SD_{Control}$ = 106.62 µm).

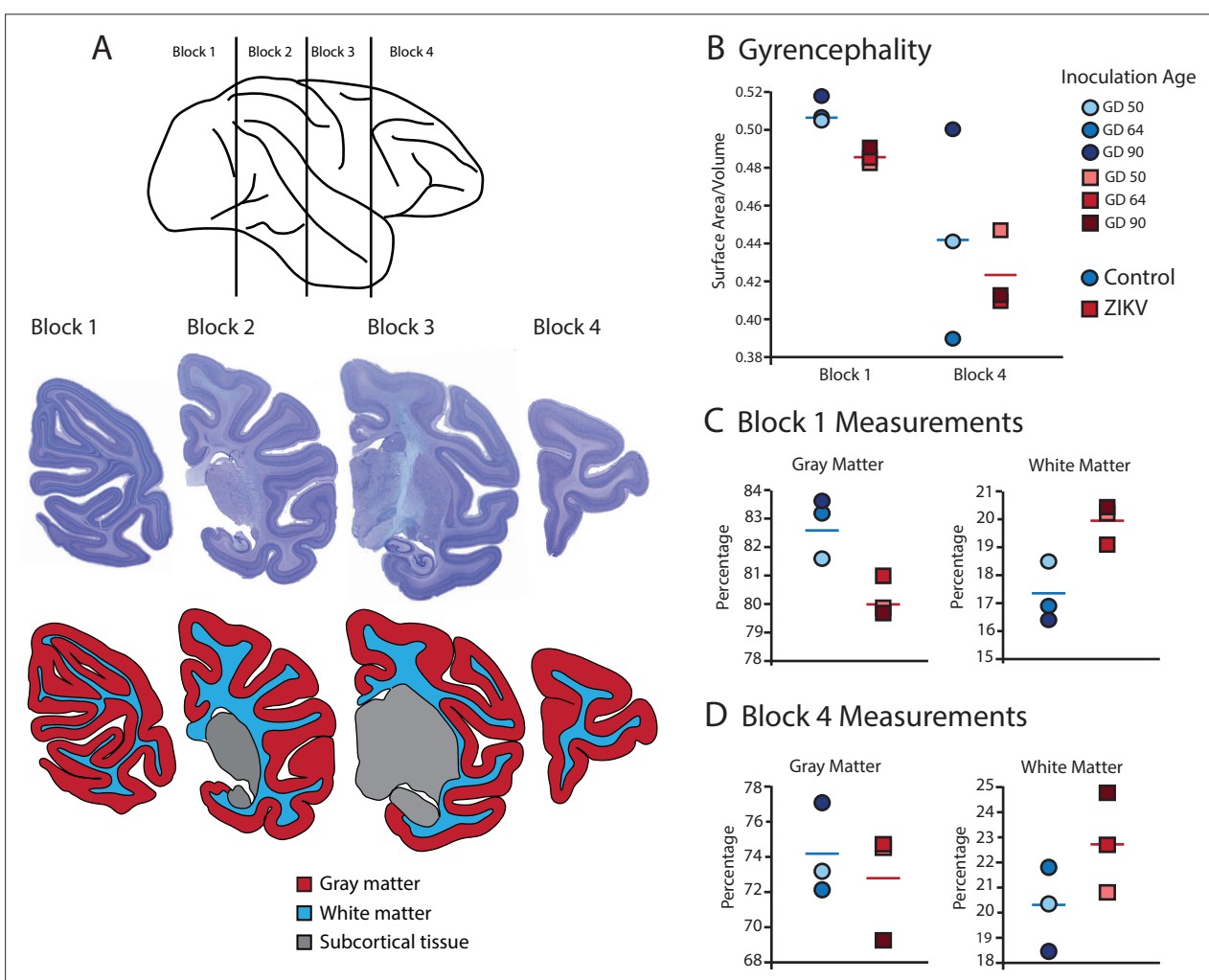

**Figure 3.** Fetal ZIKV infection results in variation in cortical development. (**A**) Each fetal brain was divided into four blocks to ensure complete rapid fixation (top). Representative Nissl-stained sections from each block (middle), and the distributions of gray matter (red), white matter (blue), and subcortical tissue (gray) are shown for each section (bottom). For (**B**)–(**D**), data from control animals are depicted in blue circles and data from ZIKV-infected animals are depicted in red squares. Gestational day (GD) of infection is indicated with different intensities of blue and red. The mean value for each group is depicted with a horizontal line (blue for control, red for ZIKV). (**B**) The gyrencephality index of the occipital lobe (Block 1; left) and frontal lobe (Block 4; right) is shown for each individual subject. Areas 17 and 46 were analyzed from Blocks 1 and 4, respectively. (**C**) Proportion of gray matter (left) and white matter (right) in Block 1. (**D**) Proportion of gray matter (left) and white matter (right) in Block 4.

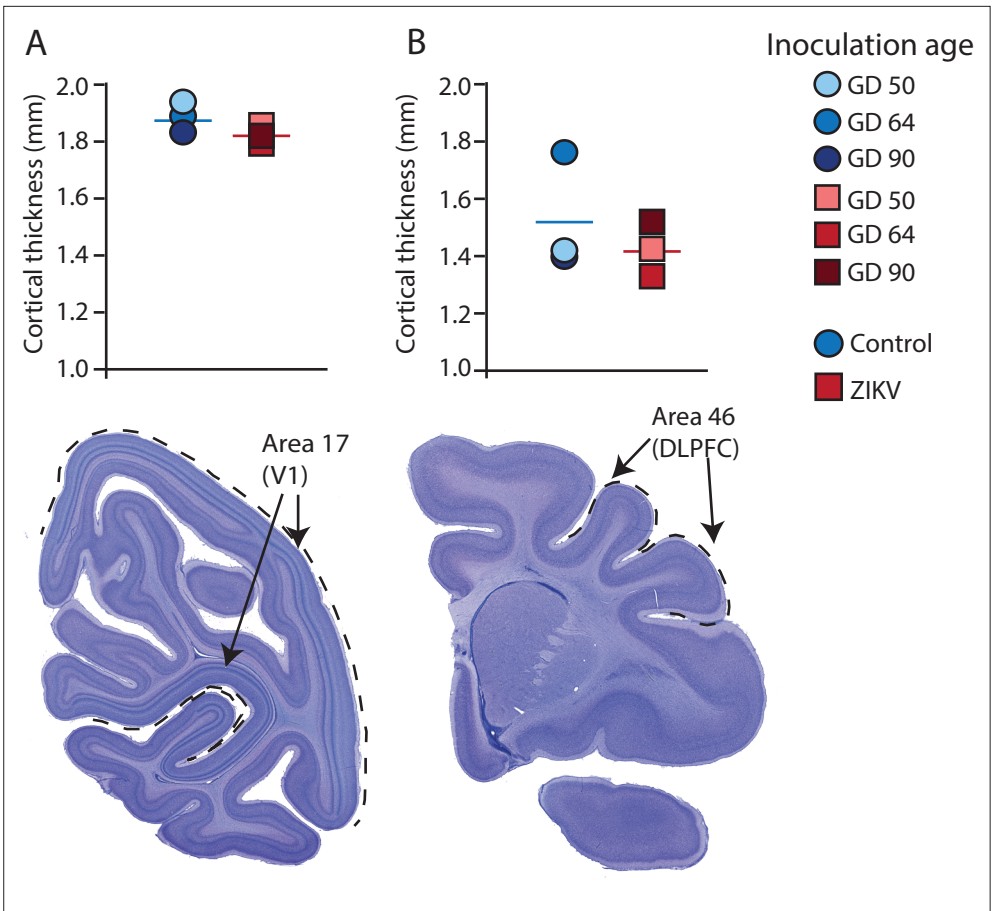

**Figure 4.** Cortical thickness does not differ across immature versus mature cortical regions of the brain in ZIKV-infected and noninfected animals. The cortical thickness of a mature cortical region (Area 17, primary visual cortex, V1) (**A**) did not differ between the control (blue circles) and ZIKV-infected (red squares) animals. The anatomical localization of Area 17 is shown in a photomicrograph of a Nissl-stained section stained (dashed lines = the extent of Area 17). (**B**) Cortical thickness of Area 46 did not differ between the control (blue circles) and ZIKV-infected animals, although there was more individual variation than was seen in Area 17. The anatomical localization of Area 46 (dorsolateral prefrontal cortex, DLPFC) is shown in a photomicrograph of a Nissl-stained section (dashed line = the extent of Area 46). Conventions as in previous figures.

Given the senior author's lab's focus on the amygdala and hippocampus, we also analyzed volumes of those structures. There were no differences between the ZIKV and control groups in the volumes of the amygdala, hippocampus. Amygdala: $t_4 = 0.095$, p = 0.93; $d = 0.078$; $Mean_{Zika} = 79.85$ mm$^3$, $SD_{Zika} = 3.88$ mm$^3$; $Mean_{Control} = 79.28$ mm$^3$, $SD_{Control} = 9.45$ mm$^3$. Hippocampus: $t_4 = 0.58$, p = 0.59; $d = 0.47$; $Mean_{Zika} = 135.82$ mm$^3$, $SD_{Zika} = 10.65$ mm$^3$; $Mean_{Control} = 143.97$ mm$^3$, $SD_{Control} = 21.84$ mm$^3$.

Finally, given the impact of ZIKV on the lateral geniculate nucleus (LGN) and observations about its impact on the retina, we also carried out a volumetric analysis of the optic tract. There were no group differences between the ZIKV and control animals in the volume of their optic tracts: $t_4 = 1.061$, p = 0.35; $d = 0.87$; $Mean_{Zika} = 31.24$ mm$^3$, $SD_{Zika} = 6.54$ mm$^3$; $Mean_{Control} = 25.09$ mm$^3$, $SD_{Control} = 7.62$ mm$^3$.

## Microstructural changes to glia

Following macrostructural evaluations, we carried out a series of IHC analyses to quantify cell-level features that might be impacted by ZIKV infection with a specific focus on glia (microglia and astrocytes) in Brodmann's Areas 17 and 46. The three-dimensional space glia occupy, especially microglia, change constantly, because these cells are highly dynamic and retract and expand their processes in response to subtle changes in the surrounding environment. Several studies have shown that glial cells have a strong form-to-function mapping and investigating morphological alterations in these

cells allows inferences about their activation state and their inflammatory status across different brain regions (*Karperien et al., 2013*; *Morrison and Filosa, 2013*; *Young and Morrison, 2018*). For example, using the combination of high-resolution confocal microscopy with three-dimensional reconstruction of individual glia volumes, we recently described how morphological changes in these cell types are connected with spine loss and neuron death in two different monkey models of Alzheimer's disease (*Seelke et al., 2020*; *Beckman et al., 2021*). Here, we use a similar approach of analysis by randomly selecting 28 cells from each anatomical area, for each animal. Iba1 and GFAP were used as general microglia and astrocytes markers, respectively, and each cell was exported and analyzed individually for total and cell body volumes, and for number of terminal point (branching ramification).

Analysis of the microglia in Area 17 revealed no significant differences in the cell body size ($t_4$ = 0.36, p = 0.74; $d$ = 0.29; *Figure 5C*). The whole cell volumes were marginally smaller in ZIKV-infected animals than in control animals and while the p value did not reach conventional levels of significance, the effect size was very large ($t_4$ = 2.71, p = 0.054; $d$ = 2.11; *Figure 5D*) There were no group differences in the number of terminal points ($t_4$ = 0.83, p = 0.45; $d$ = 0.68; *Figure 5E*). In contrast, while there were no significant differences between the cell body size of microglia in Area 46 of ZIKV-infected compared to control animals ($t_4$ = 2.07, p = 0.11; $d$ = 1.69; *Figure 5M*), ZIKV-infected animals, compared to controls, had significantly smaller whole cell volumes ($t_4$ = 9.41, p < 0.001; $d$ = 7.68; *Figure 5N*) and significantly fewer terminal points ($t_4$ = 4.18, p = 0.01; $d$ = 3.41; *Figure 5O*). The smaller whole cell volumes and terminal points suggest that the microglia were in their activated state in frontal cortex.

There were no significant group differences in astrocyte morphology in either Area 17 or 46. In Area 17, astrocyte cell body size did not differ between ZIKV-infected and control animals ($t_4$ = 0.34, p = 0.75; $d$ = 0.28; *Figure 5H*), whole cell volume ($t_4$ = 2.13, p = 0.09; $d$ = 1.81; *Figure 5I*), or number of terminal points ($t_4$ = 0.44, p = 0.68; $d$ = 0.34; *Figure 5J*). In Area 46, astrocyte cell body size did not differ between ZIKV-infected and control animals ($t_4$ = 0.10, p = 0.93; $d$ = 0.81; *Figure 5R*), whole cell volume ($t_4$ = 1.03, p = 0.36; $d$ = 0.84; *Figure 5S*), or number of terminal points ($t_4$ = 1.22, p = 0.29; $d$ = 0.99; *Figure 5T*).

## Microstructural changes related to neural development

A neuroinflammatory response, mainly driven by CD68/Iba+ microglia, in combination with the presence of ZIKV envelope protein, indicated that the frontal lobe for all ZIKV-infected subjects was a site of persistent ZIKV infection for a minimum of 60-day postinoculation. However, the consequences of the infection persist beyond the window of virus replication. Previous studies have demonstrated that ZIKV infection is associated with increased apoptosis of neural progenitor cells (*Li et al., 2016*; *Souza et al., 2016*). IHC fluorescent microscopy analyses for DAPI, cleaved caspase-3 (CC3), and SATB2, which are markers of nuclei, apoptosis onset, and immature neurons, respectively, were carried out in order to determine if the increased inflammatory response or active viral presence could induce neuronal loss across the regions analyzed 3D analysis of the average total number of cells (DAPI+), average number of immature neurons (SATB2+), and total number of apoptotic cells (CC3+), did not significantly differ between ZIKV and control animals in both regions analyzed (*Figure 6F* – Area 17, *Figure 6R* – Area 46). The number of cells in which CC3 and SATB2 was colocalized did not differ across the ZIKV-infected and control groups and the numbers were fairly low ($t_4$ = 0.65, p = 0.55; $d$ = 0.53; *Figure 5A–K*), indicating low frequency of immature neuron death. In contrast, there was significantly more frequent death of immature neurons in Area 46 of the ZIKV-infected compared to control animals, as indicated by a greater number of cells that expressed both CC3 and SATB2 ($t_{2.20}$ = 5.37, p = 0.027; $d$ = 4.39; *Figure 6L–Z*). Many CC3+ cells in the Area 46 of ZIKV animals are constituted by apoptotic immature neurons, higher resolution 3D analysis showed that activated microglia (HLA-Dr+) also present high expression of cell death marker CC3+ (*Figure 6—figure supplement 1*). The higher frequency of immature neuron death, the active immune response (activated microglia), and the presence of ZIKV protein in Area 46 suggest that frontal cortex remained a site of active ZIKV-induced neuronal remodeling at the time the brains were analyzed. Fragmented DNA was also observed in this area (*Figure 6—figure supplement 2*).

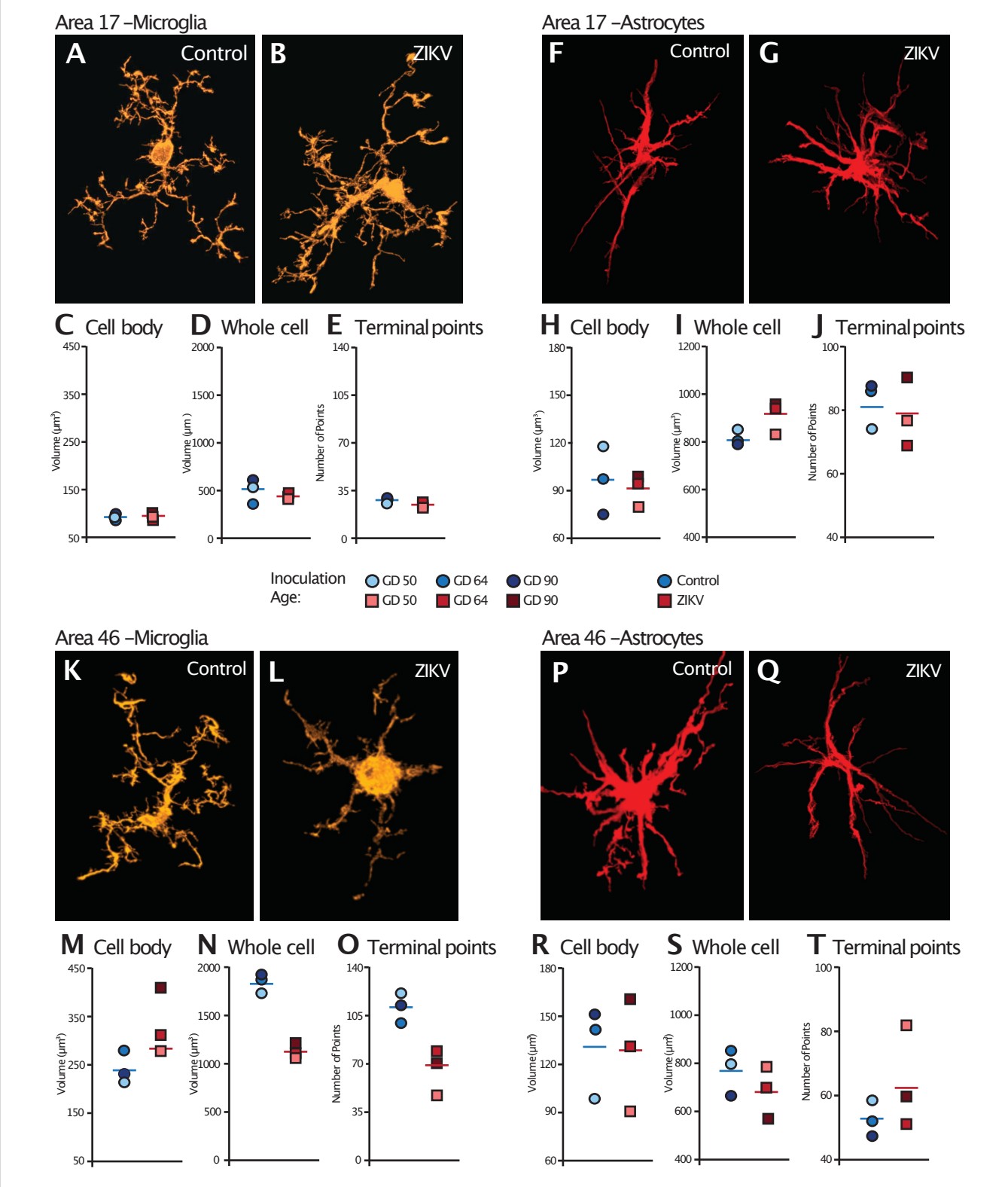

**Figure 5.** Glial cell morphological changes induced by ZIKV infection of the brain. For each cell type (microglia and astrocytes), 28 cells were randomly selected from Area 17 and from Area 46 for each animal. 3D confocal images of each individual cell were exported to Imaris software and analyzed for cell body and whole cell volume, and total number of terminal points. Representative microglia from Area 17 in control and ZIKV-infected animals are depicted in (**A**) and (**B**). The average for each condition analyzed is represented for each animal in both groups. There were no group differences in cell

*Figure 5 continued on next page*

*Figure 5 continued*

body volume (**C**), whole cell volume (**D**), or number of terminal points (**E**), a measurement for glia complexity. Similarly, there were no group differences in astrocyte anatomy (F and G for representative anatomy, H–J for data). In contrast, there were group differences in Area 46 (representative images of Area 46 microglia, K and L; astrocytes, **P and Q**). The cell bodies of ZIKV-infected animals' microglia were larger in size (**M**) than control animals. Compared to control animals' microglia, ZIKV-infected animals' microglia whole cell volumes were smaller (**N**) and had fewer of terminal points (**O**). Compared to controls animals' astrocytes, ZIKV-infected animals astrocytes tended to have smaller cell body volumes (**R**), no differences in whole cell volumes (**S**), and a greater number of terminal points (**T**).

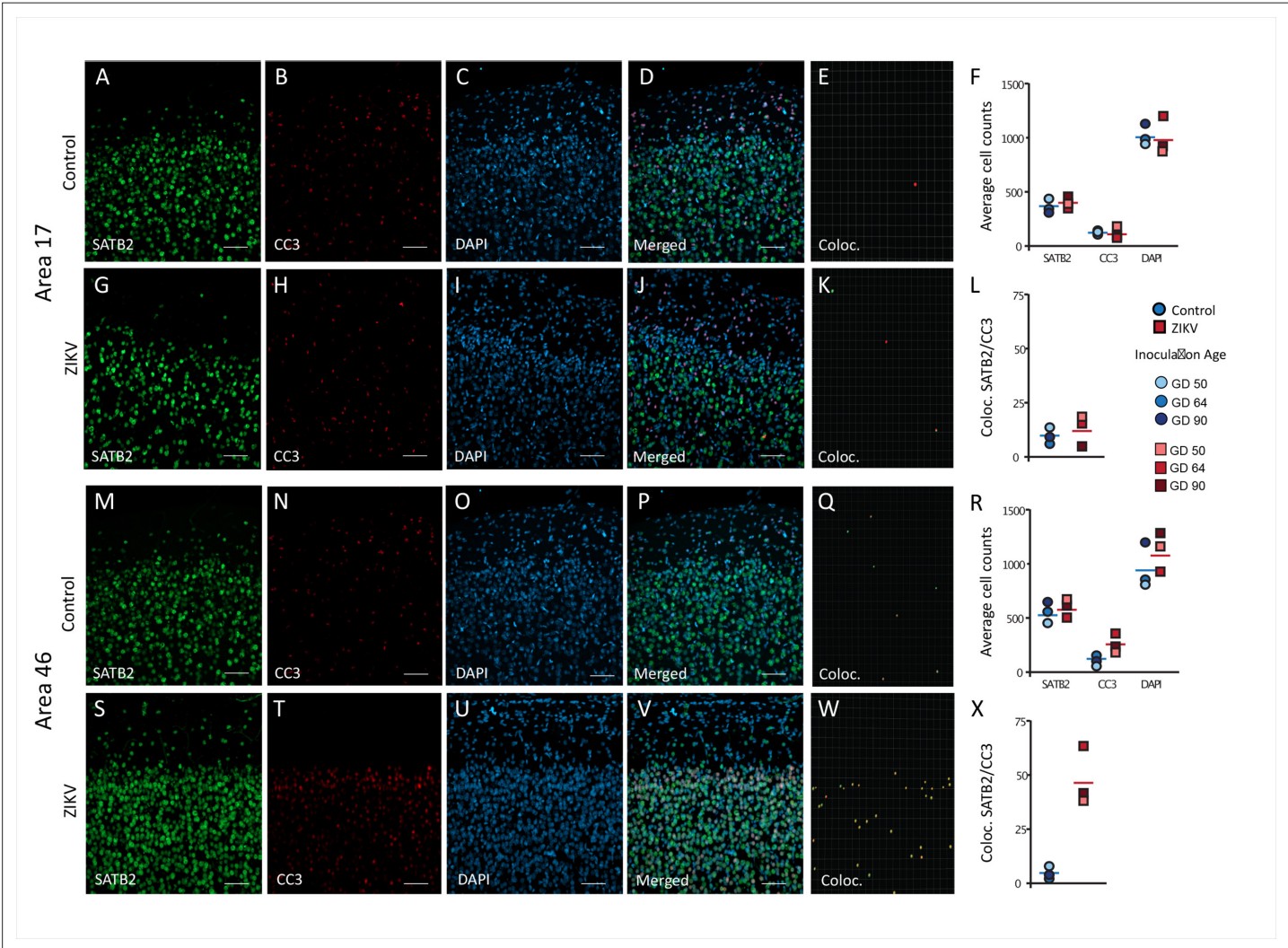

**Figure 6.** Increased death of immature neurons in Area 46 of ZIKV-infected animals. Representative tissue samples from Area 17 (**A–L**) and Area 46 (**M–Z**) in control and ZIKV-infected animals. Immature neuronal marker SATB2 (**A, G, M, S**) colocalization with CC3 (**B, H, N, T**), an apoptotic marker, and DAPI (**C, I, O, U**) were analyzed in Areas 17 and 46 of control and ZIKV-infected animals. Merged images (**D, J, P, V**). 3D surface rendering reconstruction of SATB2 and CC3 are shown in **E**, **K**, **Q**, and **X**. Average total cell counting (**F, R**) and quantification of SATB2/CC3 colocalization (**L, Z**) indicated no group differences in the number of immature neurons undergoing apoptosis in Area 17 (**L**), but a significantly greater immature neurons undergoing apoptosis in ZIKV-infected compared to control animals in Area 46 (**R**). Scale bar = 25 μm.

The online version of this article includes the following figure supplement(s) for figure 6:

**Figure supplement 1.** Active neuroinflammation is connected to increased apoptosis in the frontal lobe of ZIKV animals.

**Figure supplement 2.** Abnormal apoptotic clusters in Area 46 of ZIKV-infected animals.

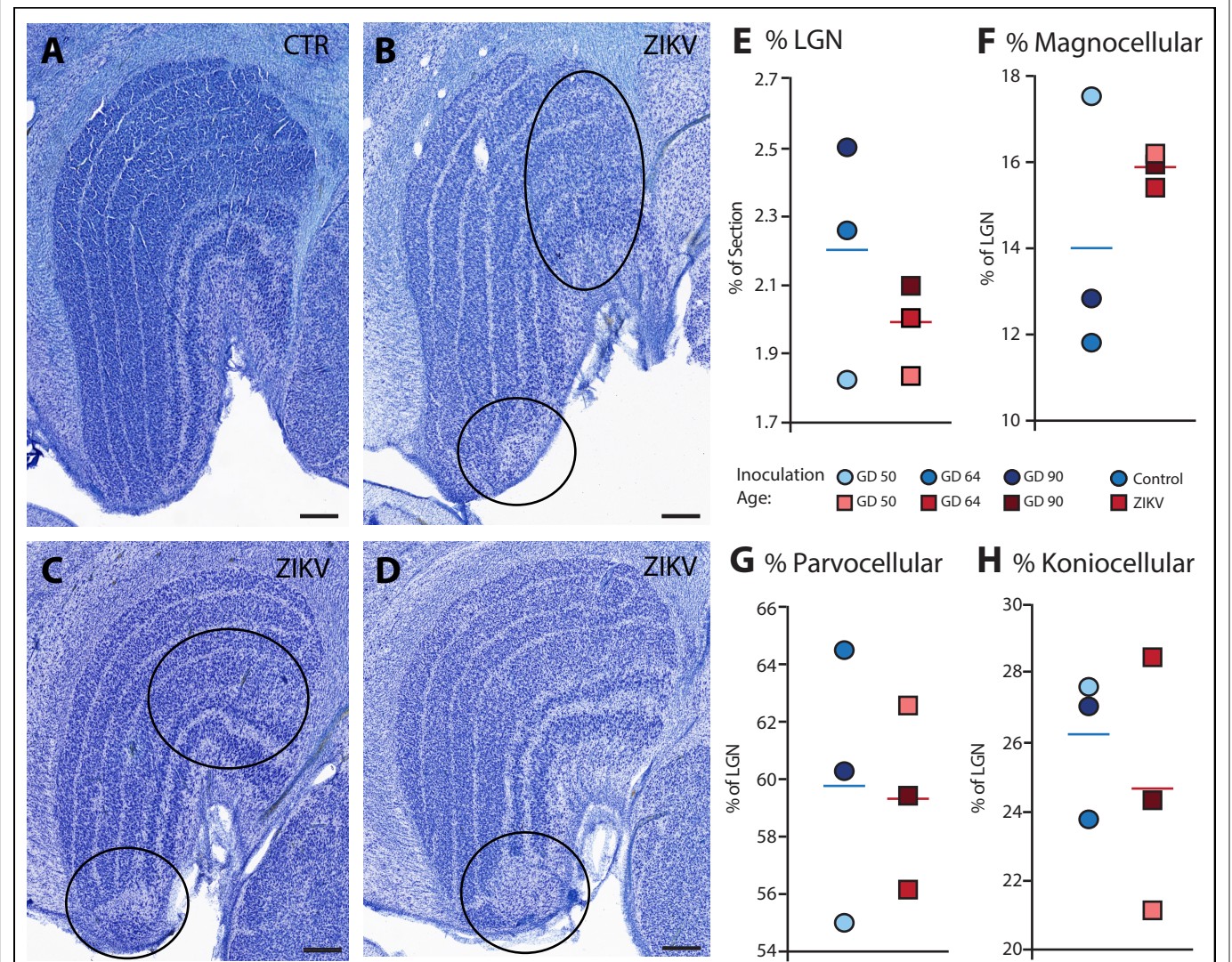

**Figure 7.** Anatomical abnormalities within the lateral geniculate nucleus (LGN) of ZIKV-infected subjects. (**A**) An example of normal caudal LGN from the control animal who was sham inoculated at GD 90. Images (**B**–**D**) illustrate LGN abnormalities from each of the three subjects infected with ZIKV. Black ovals indicate areas of anatomical abnormalities within each section. (**B**) Two instances of anatomical abnormalities within the animal infected with ZIKV at GD 50. The top oval highlights an area of undifferentiated lamination, where layers 1 and 2, and 3 appear to blend together. The bottom oval highlights an area where the normal laminar structure is interrupted by a region of low cell density. (**C**) Two instances of anatomical abnormalities within the animal infected with ZIKV at GD 64. The top oval highlights an area of undifferentiated lamination, where layers 2 and 3 appear to blend together. The bottom oval highlights an area where the normal laminar structure is interrupted by a region of low cell density. (**D**) An instance of laminar interruption of layers 1, 2, and 3 by a region of low cell density in the third ZIKV-infected animal (infected at GD 90). (**E**) The proportion of the histological section occupied by the LGN did not significantly differ between control animals and the ZIKV-infected animals, $t_4 = 1.00$, p = 0.37, $d = 0.81$. There were no group differences in the proportion the (**F**) magnocellular ($t_{2.08} = 1.02$, p = 0.36, $d = 0.84$), (**G**) parvocellular ($t_4 = 0.14$, p = 0.90, $d = 0.11$), and (**H**) koniocellular layers ($t_4 = 0.57$, p = 0.60, $d = 0.47$). Scale = 250 μm.

## Additional neuronal abnormalities

Visual inspection of the Nissl-stained sections of each brain revealed other ZIKV infection-related abnormalities within the visual pathway in addition to those observed in Area 17 (discussed above). These abnormalities included structural changes to the LGN. The LGN is a six-layered structure that is found in the ventral thalamus that receives retinal input from both eyes via the optic tract (*Jones, 1985*; e.g., *Figure 7A*). In the ZIKV-infected subjects, we observed multiple instances of laminar discontinuities within the LGN (*Figure 7B–D*). The LGNs of the control subjects were all normal.

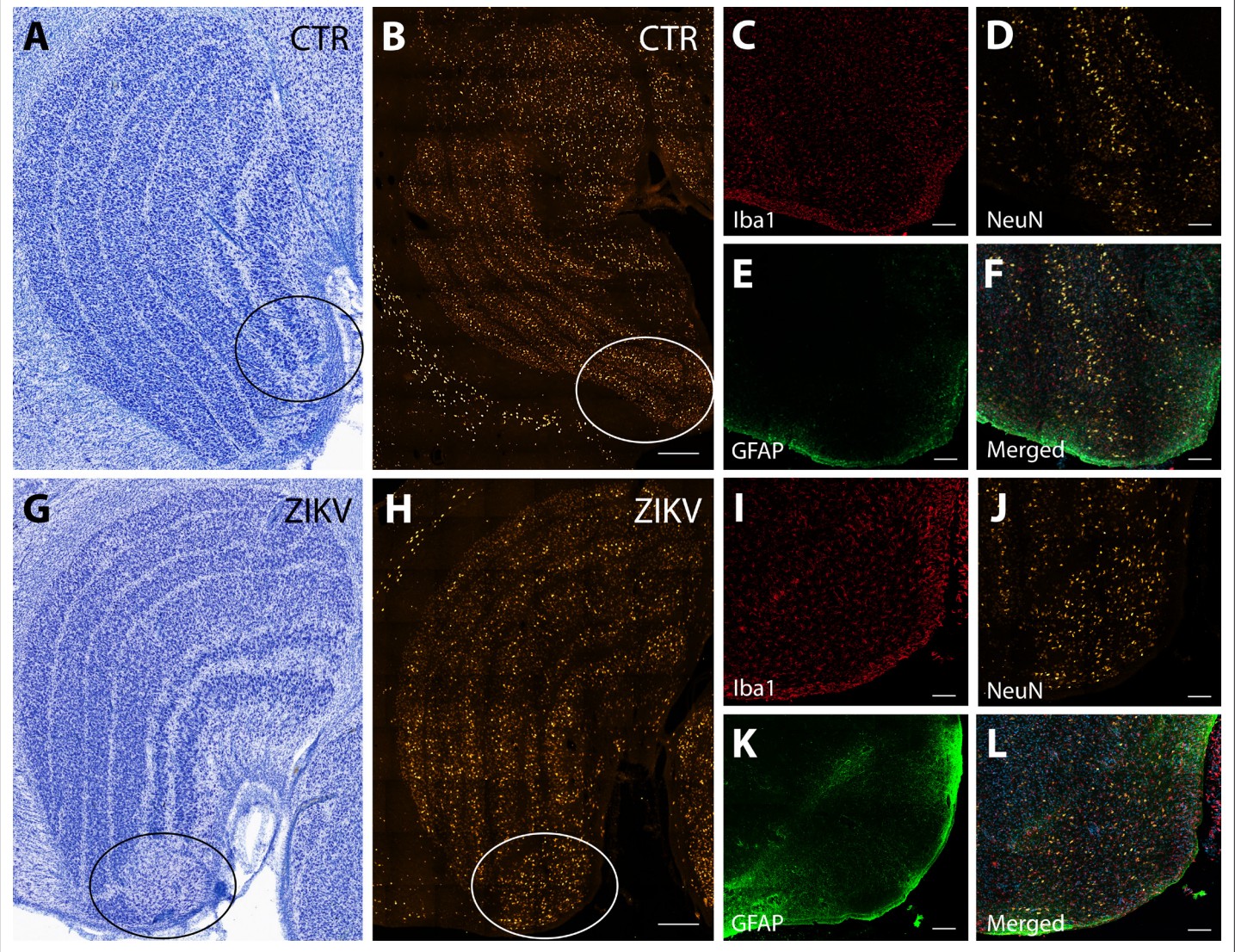

**Figure 8.** Neuroinflammatory abnormalities in the lateral geniculate nucleus (LGN) of ZIKV-infected animals. The LGNs of control animals were all normal (**A–F**), but structural abnormalities were detected in the LGNs of all three ZIKV-infected fetuses in the Nissl-stained sections (an example depicted in **G–L**). A tissue section adjacent to each Nissl stained was analyzed using confocal microscopy. Neuronal layers and glia were analyzed using NeuN (**B, H**), Iba1 (C, D; red), and GFAP (E, K; green). Zoomed NeuN images in the same area are in D and J. Merged images are found in F and L. Scale = 250 µm in G–K, scale = 25 µm in B–F and L–P, scale = 10 µm in Q–U.

The first type of LGN pathology consisted of a blurring of the expected boundaries between cell layers. In healthy tissue, the layers of the LGN are separated by a thin layer of koniocellular tissue. However, each ZIKV-infected animal had discrete instances where the koniocellular layer was absent, resulting in indistinct boundaries between layers (*Figure 7B–D*). The location of the restricted absence of the koniocellular layer varied across cases. These abnormalities occurred between both magnocelluar and parvocellular layers and could be found in both the dorsomedial and ventrolateral aspects of the structure. To investigate the differences observed in the LGN of ZIKV-infected animals, fluorescent microscopy was performed in adjacent sections of the Nissl-stained LGN (*Figure 8A–L*). In comparison to the control animals (*Figure 8B–F*), in the ZIKV-infected animals, there was an increase in the frequency and intensity of the markers for microglia (Iba1) and astrocytes (GFAP) across different layers of the LGN (*Figure 8H–L*). Although these data suggest the possibility of active inflammatory response occurring, no active virus was observed in this region. The LGN sends direct projections to layer 4 of the primary visual cortex (V1, Area 17) in the occipital lobe. We identified two categories of anatomical abnormalities in the occipital lobe of ZIKV-infected animals (but not controls):

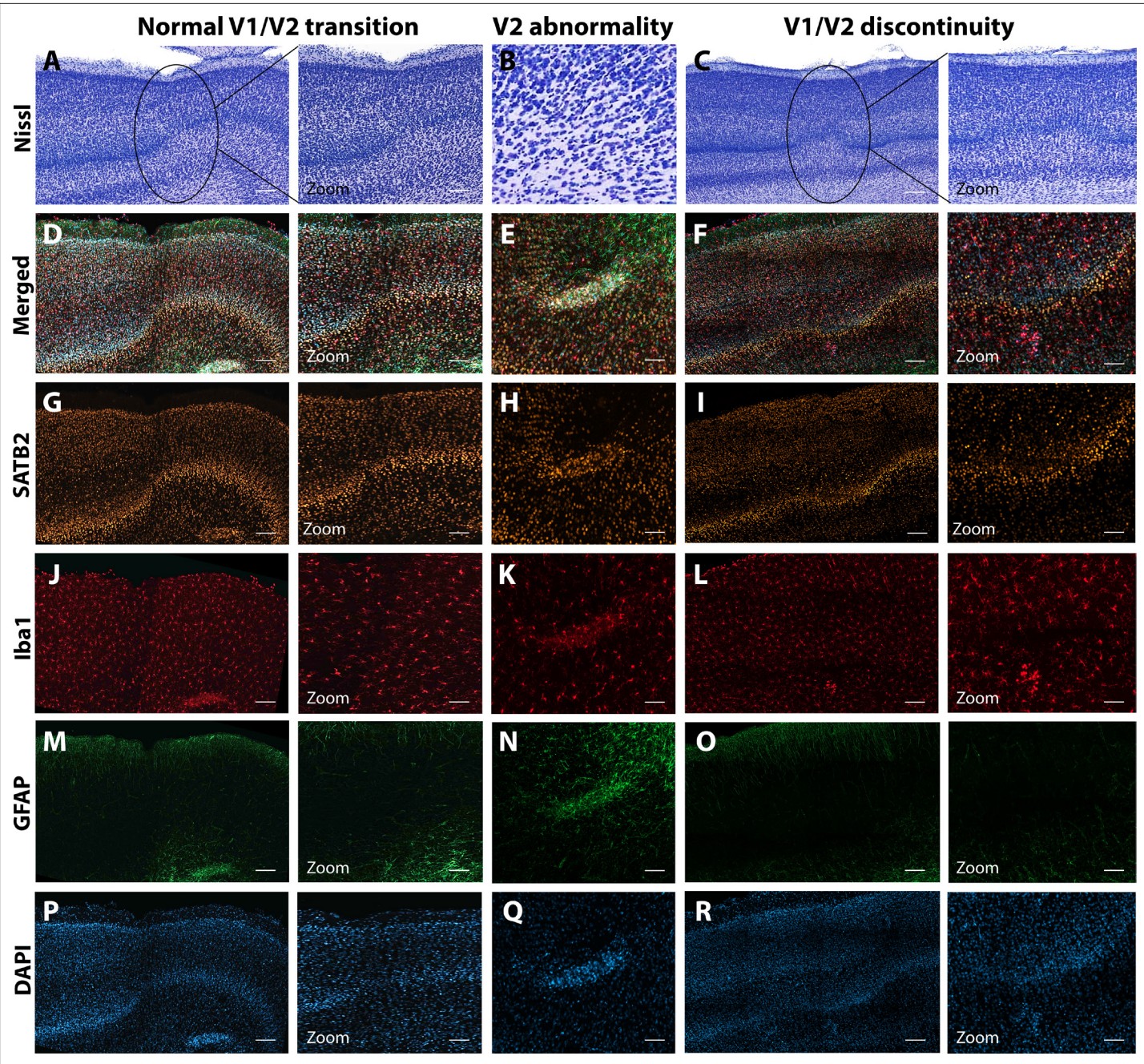

**Figure 9.** Anatomical abnormalities within the primary and secondary visual cortex of a ZIKV-infected animals. Compared to normal transitions between primary (V1) and secondary (V2) visual cortex (**A**), cellular abnormalities in the transition were detected in ZIKV-infected monkeys (**B, C**) on the Nissl-stained sections. A deeper investigation using quadruple fluorescent staining (merged images in D–F) was performed in adjacent sections for SATB2 (**G–I**), Iba1 (**J–L**), and GFAP (**M– O**). A normal transition between V1 and V2 is characterized by the uniform contraction of layer 4 and the absence of laminar discontinuity (**A, C, E, G, I, K**). We identified a region of laminar discontinuity within the boundaries of V1 (B) that appears to be the caused by a disruption to layer 4a that resulted in the displacement of layers 4b and 4c toward the pial surface (seen in **M–R**). An additional abnormality was found below layer six in area V2 (seen in **A–K** and magnified in **B–L**). Disrupted neuronal layers present are surrounded by robust presence of activated microglia and astrocytes, as highlighted in the zoom area N–R. Scale = 250 μm in A–L, scale = 25 μm in M–R.

The online version of this article includes the following figure supplement(s) for figure 9:

**Figure supplement 1.** Suprapial cortical heterotopias observed only in ZIKV-infected animals.

laminar disruptions and cortical heterotopias. Laminar disruptions were identified within Area 17 and consisted of a break in the continuity of layer 4a (*Figure 9A–R*). These disruptions appeared to be specific to layer 4a and not associated with the normal transition from Area 17 to 18, because layers 4b and 4c were still present immediately proximate to the abnormal layer 4a. This disruption is likely functionally significant, as layer 4a receives direct projections from the LGN (*Thomson and Bannister, 2003*; *Miller, 2003*), and the absence of cells that make up layer 4a likely represent lost visual information. Cortical heterotopias were identified as ectopic islands of neurons extending into cortical layer 1 (*Figure 9—figure supplement 1*).

## Discussion

Our results demonstrate that ZIKV infection during fetal development results in central nervous system abnormalities that track with temporal patterns of brain development. Importantly, our previous report on these animals (*Coffey et al., 2018*) documented evidence of maternal and fetal virus replication consistent with evidence presented here that ZIKV directly infected the brains of the fetuses. In the current study, virus replication and local neuroinflammatory response were observed in the neocortex of all three ZIKV-infected fetal brains. The types and magnitudes of these abnormalities varied across the cortex, with altered macrostructural features in the occipital cortex and microstructural abnormalities found in the frontal regions of the cortex. These regional differences likely reflect the graded manner in which the neocortex develops, with ZIKV infection tracking temporally with neural development. Importantly, the neuroanatomical abnormalities that we observed in all three ZIKV-infected fetal brains, but not in control brains, occurred even in the absence of the extreme ZIKV infection phenotype documented in human infants, microcephaly. These findings may help explain why some infants born with normal sized heads during the ZIKV epidemic have developed significant sensory, motor, and sociocognitive challenges as they grow up (*Wheeler et al., 2018*; *Cardoso et al., 2019*; *Ventura et al., 2017*; *Vianna et al., 2018*; *Marques et al., 2019*; *Pinato et al., 2018*).

The neuroanatomical abnormalities that we identified here appeared to follow the pattern of otherwise normal neuronal development. In mammals the neocortex develops in a gradient. Neural progenitor cells undergo migration in a caudal-to-rostral pattern, resulting in occipital regions having mature laminar organization before mature laminar organization is present in frontal regions (*Workman et al., 2013*; *Charvet and Finlay, 2014*). Rhesus monkeys, like humans and NHPs, have frontal cortices that do not reach maturity until well after birth (*Bourgeois et al., 1994*; *Petanjek et al., 2011*; *Mrzljak et al., 1991*). We observed macrostructural differences (e.g., relative gray/white matter volumes) in the occipital lobe which develops first and microstructural and immune-related differences in frontal lobe which develops later. Our analyses revealed evidence that microglia were in their activated state in Area 46 of ZIKV-infected but not control animals, as indicated by smaller whole cell volumes and fewer terminal points. Microglia play a critical role in neural development via a number of mechanisms (including promoting neuronal survival and removing redundant or dysfunctional synapses) and morphological differences across the groups may be evidence of pathological neuronal development. Supporting the idea that ZIKV infection interferes with neural development was our finding that ZIKV-infected fetuses had a greater number of immature cortical neurons undergoing apoptosis compared to control animals. That we found these differences in Area 46, which was still undergoing development at the time of tissue harvest, and not Area 17, which was fully laminated at the time of tissue harvest, underscores the ability of ZIKV to interfere with normative neural development. One interpretation of these data is that at the point at which the tissue was harvested, damage had already been done to the part of the brain that was differentiated but the processes of damage were underway in the tissue that was actively developing. These results suggest that infants infected with ZIKV during fetal development and born with normal sized brains may have neurodevelopmental abnormalities that only become apparent later in development. This finding is of particular importance because most infants born during the 2015–2016 ZIKV epidemic in the ZIKV hotspots were born with normal sized heads and microcephaly, while an extreme phenotype, is thought to be relatively low frequency (*Cardoso et al., 2019*; *Aragao et al., 2017*; *Faiçal et al., 2019*).

One seemingly consistent anatomical abnormality found in both human patients and animal models of ZIKV is an increased frequency of abnormalities in the visual system (*Aleman et al., 2017*; *Fernandez et al., 2017*; *Miner et al., 2016b*; *Simonin et al., 2019*; *Singh et al., 2018*; *Ventura et al., 2019*; *Ventura et al., 2016*; *Verçosa et al., 2017*), especially the retina. Patients present with

severe eye disease, including congenital cataracts, optic nerve abnormalities, chorioretinal atrophy, intraretinal hemorrhages, mottled retinal pigment epithelium, and blindness (*Ventura et al., 2017*; *Aleman et al., 2017*; *Fernandez et al., 2017*; *Ventura et al., 2019*; *Ventura et al., 2016*; *Verçosa et al., 2017*; *Miner et al., 2016a*; *Ventura et al., 2018*). Further, there is some evidence that ZIKV can be transmitted from the retina to the rest of the visual system via the optic nerve, although this has yet to be confirmed in humans or NHPs (*van den Pol et al., 2017*). Anatomical abnormalities that we observed in the LGN and Area 17/V1 are consistent with infection and developmental disruption of the eye and optic nerve, insofar as projections from the retina enervate the LGN in a retinotopic manner, and projections from the LGN in turn enervate the primary visual cortex (Brodmann's Area 17) (*Shatz, 2018*; *Rakic, 1977*). The anatomical and functional retinotopic organization of both LGN and V1 is dependent upon activity during developmental critical periods (*LeVay et al., 1980*; *Ghosh and Shatz, 1994*). Insults during early development that disrupt this activity can result in damage to brain regions downstream of the initial insult (*Ito et al., 2008*; *Eysel and Wolfhard, 1984*; *Yücel et al., 2003*). The lesions and laminar dysregulations observed in the LGN and V1, respectively, have significant functional consequences.

While routine H&E histopathology of the eyes did not reveal any significant lesions in these animals (*Coffey et al., 2018*), subsequent work from our group has identified retinal pathology in one infant that was infected with ZIKV IA as a fetus and monitored for 2 years after birth (*Yiu et al., 2020*). The retinal tissue was used for pathological analysis and was unavailable for use in this study. However, the retinotopic organization and activity-dependent development of LGN layers suggests that the dysregulation we see within the LGN is likely related to lesions of the retina caused by the Zika virus. Future studies would benefit from examination of the entire visual pathway, from retina to LGN to V1, and visual association regions targeted by V1.

While our sample size was small ($N$ = 3 in each group), only one hemisphere of the brain was available for our analyses, and the timing of ZIKV infection varied across animals (with one ZIKV and one procedure-matched control at each of three infection dates, GD 50, 64, or 90), all three infected subjects showed the same pattern of pathology: active inflammation in the still-developing regions of the frontal cortex paired with anatomical abnormalities in the already-developed regions of the caudal cortex. Additionally, each infected fetus had evidence of active virus replication as late as 105-day postinoculation, which could explain the ongoing neurologic damage. Although the sample size here is not atypically small for neuroanatomical analyses of NHP tissues or initial infectious diseases studies of this sort, it is small and thus important to note that we carried out both experimental and statistical procedures to address potential sample size concerns. First, each infected animal was paired with a procedure-matched control that underwent identical procedures without viral inoculation. Second, all conducted statistical analyses are reported (i.e., there was no p-hacking). Third, raw data but not statistics are displayed in figures encouraging the reader to draw conclusions from the patterns of data within and across groups rather than relying solely on p values. While replication in a larger sample is a laudable future goal, the present study also sets the stage for targeted hypothesis testing about the timing of infection and gestational development which seems to matter less in the case of ZIKV than potentially anticipated.

Previous studies from our group (*Coffey et al., 2018*) and others (*Adams Waldorf et al., 2018*; *Martinot et al., 2018*; *Mohr et al., 2018*; *Steinbach et al., 2020*) have identified discrete histopathological abnormalities in the brain tissue of macaque fetuses infected with ZIKV, including calcifications with gliosis and a loss of neural precursor cells. In addition to previous neuropathological evaluations, previous evaluation of the other hemisphere of each of these brains identified ZIKV RNA in Area 46 in all infected animals, and in occipital pole (Area 17) of animals infected at GD 64 and 90 (*Coffey et al., 2018*). The histological analyses presented here differ in that we measured specific cytoarchitectonic regions of the brain as well as morphological analyses of glial cells in the frontal and occipital cortex. Together, these analyses paint a picture of inflammatory processes related to viral infiltration sweeping through the cortex and a wave of cell death. Given our analyses, it is not possible to tell whether inflammatory processes are caused by or are causing cell death, but they certainly appear to co-occur given their proximity. Importantly, in addition to underscoring the importance of adopting quantitative neuroanatomical analyses to understand viral infections of the developing brain, these findings underscore the importance of using a monkey model because we would not expect to see such patterns in nonprimate models,

given that rodents do not have a homolog of Area 46 and a significant period of their brain development occurs after birth.

While we focus heavily on the importance of homologs between humans and rhesus monkeys both in the course of ZIKV infection and the development of PFC (including Area 46), the rhesus monkey model opens the possibility of linking neural development to tractable, translatable behavioral development, and also developing translatable interventions to improve neural and behavioral function. Rhesus monkeys are a robust model for human infant development across a variety of behavioral domains – including cognition, affect, and social behavior (*Phillips et al., 2014*; *Clancy et al., 2013*; *Fox, 1984*; *Cassidy and Shaver, 1999*). Moving forward, this will allow us to understand how the neural abnormalities associated with fetal ZIKV infection, such as those documented here, impact infant behavioral development. Because rhesus monkeys develop three to four times faster than humans (*Workman et al., 2013*), a separate, currently ongoing cohort of infants who were infected with ZIKV as fetuses will soon out-age human infants who were infected in utero, thus providing a prospective look at the challenges those infants may face. In this context, our long-term developmental studies which have documented significant neural and behavioral plasticity following early brain damage (e.g., *Grayson et al., 2017*; *Bliss-Moreau et al., 2017*) provide hope that fairly simple behavioral interventions may be possible to improve the outcomes for impacted children. This is particularly important given that the ZIKV epidemic disproportionally impacted people of color living in socioeconomically disadvantaged communities (*Moreno-Madriñán et al., 2017*; *Kapiriri and Ross, 2020*; *Gómez et al., 2018*) and that ZIKV is expected to return seasonally in the future (*Tesla et al., 2018*).

## Materials and methods

All procedures were approved by the University of California, Davis Institutional Animal Care and Use Committee (Protocol # 19211) which is accredited by the Association for Assessment and Accreditation of Laboratory Animal Care International (AAALAC). Animal care was performed in compliance with the 2011 *Guide for the Care and Use of Laboratory Animals* provided by the Institute for Laboratory Animal Research.

### Study design

#### Subjects

All adult female rhesus macaques (*Macaca mulatta*) enrolled in the study were from the conventional breeding colony and were born at CNPRC. Animals were determined to be free of West Nile virus (WNV) via simian WNV ELISA (Xpress Bio), SIV-free, type D retrovirus-free, and simian lymphocyte tropic virus type one antibody negative. Animals were selected from the timed breeding program based on the date of their mating and ultimately the GD of their fetus determined by comparing data obtained on ultrasound to CNPRC colony data for fetal development. All animals had at least one and up to six previous successful pregnancies with live births. ZIKV-infected animals were enrolled first, followed by control animals. The animals were housed indoors in standard stainless-steel cages (Lab Products, Inc) that met or exceed the sizing required by NIH standards, in rooms that maintained 12 hr light/dark cycles (0600 lights on to 1800 lights off), with temperature controlled between 65–75°F and 30–70% humidity. Monkeys had free access to water and received commercial chow (high protein diet, Ralston Purina Co.) twice a day and fresh produce multiple times a week. All but one female was paired with a compatible social partner.

#### Experimental infection with Zika virus

Four pregnant females and their fetuses were inoculated with a 2015 Brazilian virus isolate (strain Zika virus/*H. sapiens*-tc/BRA/2015/Brazil_SPH2015; genbank accession number KU321639.1) by injecting 5.0 log10 PFU in 1 ml RPMI-1640 medium both IV and IA (*Coffey et al., 2018*). Each animal was inoculated only once at either 41, 50, 64, or 90 GDs. The fetus of the female inoculated at GD 41 was found dead 1 week later and its brain was autolyzed (*Coffey et al., 2018*). Three mother–fetus pairs were sham inoculated with 1 ml of RPMI-1640 medium both IV and IA at GD 50, 64, and 90 to serve as procedure-matched controls.

No formal sample size estimation was carried out because this was an initial pilot study to determine whether rhesus monkey fetuses could be infected with ZIKV (*Coffey et al., 2018*). When the fetus inoculated at GD 41 died, we did not include a control fetus with a sham inoculation at GD 41, resulting in the final sample of *N* = 3 in each experimental group.

## Histological procedures

### Tissue harvesting and fixation

As experimentally planned, the fetuses were harvested at GD 155 via fetectomy and immediately euthanized for detailed tissue collection for five of six pairs. The fetus that was infected at GD 64 was born alive at approximately GD 151 and euthanized the same day for tissue collection. Only evaluation of fetal tissues is reported here.

The left hemisphere of each brain was used for viral load determination or fixed in 10% buffered formalin for histology as described earlier (*Coffey et al., 2018*). The right hemisphere was fixed in 4% paraformaldehyde in 0.1 M sodium phosphate buffer for an hour, cut into four blocks to allow sufficient rapid penetration of the fixative into the tissues, and then fixed for 48 hr at 4°C with rotation. Blocks were numbered 1–4, starting at the caudal extent of the brain (Block 1, including the occipital lobe) and ending at the rostral extent of the brain (Block 4, including the frontal pole). The tissue was cryoprotected in 10% glycerin with 2% DMSO in 0.1 M sodium phosphate buffer overnight then 20% glycerin with 2% DMSO in 0.1 M sodium phosphate buffer for 72 hr (*Rosene et al., 1986*). Blocks were then frozen in isopentane following standard laboratory procedures (*Bliss-Moreau et al., 2017*).

## Tissue sectioning

The tissue was sectioned on a sliding freezing microtome (Thermo Scientific Microm HM430, Waltham, MA, USA) into eight series (7 at 30 μm and 1 at 60 μm). 30 μm tissue sections were placed in a cryoprotectant solution and stored at −20°C. The 60 μm tissue sections were postfixed for 2 weeks in 10% formalin and stored at 4°C.

### Nissl staining

The 60 μm sections were mounted on gelatin subbed slides and Nissl stained using 0.25% thionin (according to our standard protocols [*Lavenex et al., 2009*; *Bliss-Moreau et al., 2017*]). Slides were coverslipped using DPX mounting medium (Millipore Sigma, St. Louis, MO, USA). These sections were scanned (TissueScope LE; Huron Digital Pathology; St. Jacobs, ON, Canada) and digital images were used for analyses. Each image was coded to keep evaluators blind to experimental condition. Anatomical boundaries were determined by comparing Nissl-stained tissue to reference atlases (*Saleem and Logothetis, 2012*; *Rohlfing et al., 2012*), and all anatomical analyses were performed using StereoInvestigator (MBF Bioscience, Williston, VT, USA).

### Immunohistochemistry

30-μm-thick free-floating sections were incubated in an antigen retrieval solution (Wako, S1700) at 60°C for 30 min. Sections were incubated in blocking solution: 5% donkey serum, 5% goat serum, 5% bovine serum albumin in phosphate-buffered saline (PBS) 0.3% Triton for 2 hr at room temperature (RT) under agitation. Sections were then incubated overnight with antibodies to Iba1 (Wako, code 019-19741, 1:500), NeuN (Synaptic Systems, code 266-004, 1:1000), GFAP (Abcam, code ab4674, 1:1000), antiflavivirus group (Millipore, code MAB10216, 1:400), nonphosphorylated neurofilament (SMI32, Biolegend, code SMI-32P, 1:500), CC3 (Cell Signaling, code 9661, 1:500), CD68 (Abcam, code ab53444, 1:300), HLA-Dr (ThermoFisher, code MA5-11966), and SATB2 (Cell Signaling, code 39229, 1:500). Tissue was washed thoroughly with PBS and incubated with Alexa Fluor secondary antibodies (Invitrogen, 1:500) for 2 hr, at room temperature. Finally, sections were incubated with DAPI for 10 min after washing the secondary antibodies. Sections were mounted on microscope slides and coverslipped with Prolong Diamond Antifade (Invitrogen).

## Neuroanatomical evaluations

We first reviewed each Nissl slide from each case to qualitatively identify potential differences across brains. Given hypotheses about ZIKV's particular impact on developing neurons, we elected to quantitatively compare Areas 17 and 46, delineated from Blocks 1 and 4, respectively, because of the

caudal–rostral developmental gradient of cortical development but also report evaluations of cortical and subcortical structures and those that appeared to be compromised by ZIKV infection (e.g., LGN). Macrolevel features of the tissue were quantified on the Nissl sections (including gyrencephalization, cortical thickness, etc., described below). Finally, a series of IHC analyses on targeted regions of tissue were completed. All people analyzing tissue were blind to condition.

### Analysis of Nissl-stained tissues

To compute an index of gyrencephalization, we measured the area and perimeter of each section of tissue in StereoInvestigator. These areas were chosen because the cortex develops in a caudal to rostral pattern (*Charvet and Finlay, 2014*; *Rakic, 2002*; *Colby et al., 2011*); brain development at the time of tissue harvest should have been most complete in the occipital lobe (caudal) and least complete in the frontal lobe (rostral). Thus, if fetal ZIKV infection disrupted development in a timing specific way, we would expect to see the greatest difference in neuroanatomical patterning when these lobes were compared. The distance between each tissue section (section thickness + intersection interval) was then used to determine the surface area and volume of each section as follows:

$$\text{Surface area} = ((\text{perimeter of section } 1 + \text{perimeter section } 2)/2) * \text{intersection interval}$$

$$\text{Volume} = ((\text{area of section } 1 + \text{area of section } 2)/2) * \text{intersection interval}$$

Surface area and volume values were summed for each subject to determine the total surface area and total volume of the frontal and occipital lobes. The gyrencephality ratio for both lobes was computed by dividing the surface area (in $mm^2$) by the volume (in $mm^3$).

Gray matter and white matter volumes were computed on the same Nissl-stained sections in the occipital and frontal lobes. Gray matter was defined as cortical areas that exhibited clear lamination; white matter was defined as fiber tracks adjacent to the cortex. The outline of the section, areas of gray matter, and areas of white matter were traced in StereoInvestigator. Subcortical tissue, including the caudate and putamen and the rostral extent of the hippocampus, was present in a few sections, but these areas were not included in this analysis. The total area and volume of gray and white matter for each section were computed (as above). The proportion of each lobe that was composed of white matter and gray matter was computed by dividing the total volume of gray and white matter in each lobe by the total volume of the lobe. The occipital lobe included no subcortical structures and so the total area was occupied by either gray or white matter. The frontal lobe included subcortical structures and so the percentage of gray matter plus the percentage of white matter did not sum to 100.

Cortical thickness was computed for three areas: Brodmann's Area 17 (primary visual cortex, V1), Brodmann's Area 4 (primary motor cortex, M1), and Brodmann's Area 46 (dorsolateral PFC). Each region was chosen because of its connection with a sensory input or motor output that is known to be negatively impacted by ZIKV infection and because Area 17 is in Block 1 at the caudal extent of the brain and Area 46 is in Block 4 at the rostral extent of the brain. Humans with CZS have been documented to have vision deficiencies (*Ventura et al., 2017*) as well as muscle spasms and hypertonia (*Pessoa et al., 2018*), implicating V1 and M1, respectively. Similarly, some evidence exists that children with CZS have symptoms consistent with intellectual developmental disorders and executive function disorders that implicate Area 46 (*Cardoso et al., 2019*; *van der Linden et al., 2016*).

Boundaries of the cortical regions of interest were identified through patterns of cortical lamination and other architectonic landmarks (*Saleem and Logothetis, 2012*; *Rohlfing et al., 2012*). Cortical thickness measurements were taken perpendicular to the pial surface. In Areas 17 and 4, three thickness measurements were taken from each of three adjacent sections. In Area 46, four thickness measurements were taken from each of two adjacent sections. Measurements were first averaged within a section, then between sections, for each subject.

Four sections from each subject that included the full LGN were evaluated. This procedure was required because the brains were blocked through the LGN in five of our six subjects. For each subject, the four largest complete sections were selected from the available complete sections. Total LGN area was measured on the Nissl-stained sections in StereoInvestigator and then the area of magnocellular (layers 1 and 2) and parvocellular (layers 3–6) was measured. Koniocellular area for each section was computed as the total area minus the magnocellular and parvocellular areas.

## Analysis of tissues prepared via IHC procedures

All analyses of fluorescently stained tissue were conducted using a Zeiss LSM 800 confocal microscope (Carl Zeiss AG, Oberkochen, Germany).

## Microglia and astrocyte quantification

For microglia and astrocyte morphological analyses, 28 microglia and 28 astrocytes were selected randomly from Areas 17 and 46 of each animal. For each cell, a z-stack at ×63 magnification was made, and the image was exported to Imaris software (Oxford Instruments, Bitplane Inc, Concord, MA, USA) to create 3D volume surface rendering. Cell body volume, whole cell volume, and terminal points were quantified for each cell and averaged within each subject.

## Total DAPI, SATB2, and CC3 counting and SATB2 colocalization with CC3

Three z-stack images at ×20 magnification were obtained within both Areas 46 and 17 for each subject. Images that contained both immature neurons (SATB2) and apoptotic cells (CC3) were exported to Imaris software to quantify colocalization.

## DNA fragmentation

Apoptotic cells were analyzed using DeadEnd Fluorometric TUNEL System, from Promega (cat no. G3250). Four slides from each animal were incubated with equilibration buffer for 10 min. Slides were then incubated with TUNEL reaction mix for 60 min at 37°C in a humidifier chamber, avoiding exposure to light. The reaction consists in a catalytical incorporation of fluorescein-12-dUTP at 3-OH DNA end using Terminal Deoxynucleotidyl Transferase, forming a polymeric tail using the TUNEL principle. The reaction was stopped using SSC (NaCl and sodium citrate) and slides were washed three times with PBS, incubated with DAPI solution for 10 min and then mounted using Vectashield (Vector Lab). For each animal, one slide was incubated with RQ1 RNase-Free DNase (Prometa cat no. M6101) used as a positive control. Three images from each area analyzed were acquired at ×20 magnification for each slide for each animal.

### Data analysis

Data were analyzed in SPSS 26.0 (IBM Corp). Data were checked for normality via the Kolmogorov–Smirnov and Shapiro–Wilk tests. Data were then analyzed using independent sample t-tests with experimental group as the between subject variable. Cohen's d was computed as a measure of effect sizes. No individual data points were excluded from analysis (e.g., because they were outliers) and all conducted analyses are reported in the main or supplementary text. We elected to report each individual animals' data and not to report statistics on the figures because the sample size is small.

### Acknowledgements

Thank you to Jennifer Watanabe, Jodie Usachenko, Christina Cruzen, Kari Christe, Rachel Reader, Wilhelm von Morgenland, Anil Singapuri, Gilda Moadab, and members of the Bliss-Moreau Laboratory who contributed to this work. R21NS104692, R01HD096436, and the Murray B Gardner Junior Faculty Research Fellowship from the UC Davis Center of Comparative Medicine to EBM; R21AI129479 and a pilot grant from the California National Primate Research Center via P51OD011107 to LLC.

## Additional information

### Funding

| Funder | Grant reference number | Author |
| --- | --- | --- |
| National Institute of Neurological Disorders and Stroke | R21NS104692 | Eliza Bliss-Moreau |
| National Institute of Allergy and Infectious Diseases | R21AI129479 | Lark LA Coffey |

| Funder | Grant reference number | Author |
| --- | --- | --- |
| California National Primate Research Center | Pilot Grant P51OD011107 | Lark LA Coffey |
| Eunice Kennedy Shriver National Institute of Child Health and Human Development | R01HD096436 | Eliza Bliss-Moreau |
| National Institute of Allergy and Infectious Diseases | R21 AI129479 | Koen KA Van Rompay |
| University of California, Davis | Murray B. Gardner Jr. Fellowship | Eliza Bliss-Moreau |

The funders had no role in study design, data collection, and interpretation, or the decision to submit the work for publication.

## Author contributions

Danielle Beckman, Formal analysis, Investigation, Methodology, Visualization, Writing – original draft, Writing – review and editing; Adele MH Seelke, Formal analysis, Investigation, Visualization, Writing – original draft, Writing – review and editing; Jeffrey Bennett, Conceptualization, Data curation, Formal analysis, Investigation, Methodology, Visualization, Writing – original draft, Writing – review and editing; Paige Dougherty, Investigation, Methodology, Writing – review and editing; Koen KA Van Rompay, Conceptualization, Formal analysis, Funding acquisition, Investigation, Methodology, Project administration, Writing – review and editing; Rebekah Keesler, Investigation, Writing – review and editing; Patricia A Pesavento, Conceptualization, Investigation, Methodology, Writing – review and editing; Lark LA Coffey, Conceptualization, Formal analysis, Funding acquisition, Investigation, Methodology, Project administration, Validation, Writing – review and editing; John H Morrison, Conceptualization, Methodology, Writing – review and editing; Eliza Bliss-Moreau, Conceptualization, Data curation, Formal analysis, Funding acquisition, Investigation, Methodology, Project administration, Resources, Supervision, Visualization, Writing – original draft, Writing – review and editing

## Author ORCIDs

Adele MH Seelke http://orcid.org/0000-0002-2867-0335
Jeffrey Bennett http://orcid.org/0000-0002-3255-5663
Koen KA Van Rompay http://orcid.org/0000-0002-7375-1337
Eliza Bliss-Moreau http://orcid.org/0000-0002-0740-5612

## Ethics

All procedures were approved by the University of California, Davis Institutional Animal Care and Use Committee (Protocol # 19211) which is accredited by the Association for Assessment and Accreditation of Laboratory Animal Care International (AAALAC). Animal care was performed in compliance with the 2011 Guide for the Care and Use of Laboratory Animals provided by the Institute for Laboratory Animal Research.

## Decision letter and Author response

Decision letter https://doi.org/10.7554/eLife.64734.sa1
Author response https://doi.org/10.7554/eLife.64734.sa2

# Additional files

## Supplementary files

• Transparent reporting form

## Data availability

Data is available on OSF.

The following dataset was generated:

| Author(s) | Year | Dataset title | Dataset URL | Database and Identifier |
|---|---|---|---|---|
| Bliss-Moreau E | 2022 | Neuroanatomical abnormalities in a nonhuman primate model of congenital Zika virus infection | https://osf.io/9gdqw/ | Open Science Framework, 9gdqw |

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
