## [Editor Report]

This rigorous study provides compelling evidence that Zika virus infections in infants can markedly impact brain development through neuroinflammatory mechanisms. The work will have broad interest among developmental neurobiologists, as well as scientists whose work focuses on ZIKV pathogenesis.

---

## [Decision Letter]

**Decision letter after peer review:**

Thank you for submitting your article "Neuroanatomical abnormalities in a nonhuman primate model of congenital Zika virus infection" for consideration by *eLife*. Your article has been reviewed by 3 peer reviewers, including Rebecca Shansky as Reviewing Editor and Reviewer #1, and the evaluation has been overseen by Michael Taffe as the Senior Editor.

The reviewers all agree that the work is novel, timely, and important. Reviewers 2 and 3 had a number of issues that need to be resolved before they consider the manuscript suitable for publication. Some involve additional immunohistochemical analyses or re-analysis of existing images (Reviewer 3). Please review these comments carefully in deciding whether to submit a revised version of this manuscript.

Summary:

This paper will be of interest to those that study the neuropathological effects of viral replication on brain development in pediatric populations and for those aiming to understand ZIKV pathogenesis. A major strength is the detailed histopathological analyses on the developing brain of non-human primates that include rigorous age-matched and procedure matched-controls.

*Reviewer #1:*

Children born during the Zika outbreak are known to have marked changes in neurological function and in some cases exhibit microcephaly, but the anatomical bases for these deficits is unknown. The goal of this study was to identify neuropathologies in near-term monkeys whose mothers were infected with Zika (ZIKV) during gestation. Using a combination of histological and immunofluorescent techniques, the authors demonstrate that compared to control monkeys, the ZIKV-infected fetuses exhibit a number of anatomically discrete abnormalities that are accompanied by evidence of localized neuro-immune responses and cell death. These abnormalities are shown through a comprehensive series of fluorescent micrographs that make the differences between the infected and control monkeys clear to the reader. The quantification methodologies are explained in detail, and the authors take care to note statistical considerations for the small sample size. Importantly, the observations map onto areas whose functions are affected in reports of children infected with ZIKV in utero, and therefore this work could have a major impact on public health.

No recommendations, this is a very nicely-prepared paper that needs no improvements.

*Reviewer #2:*

This is an excellent pathological description of neuropathological changes associated with congenital ZIKV infection in the non-human primate model. Major findings include regional variation in apoptosis, neuronal and glial morphology, and structural changes in nuclei of the visual system only seen in ZIKV challenged infants. The inclusion of three procedure control animals is exceptional and although the sample numbers are small the inclusion of these controls have uncovered clear pathological features of ZIKV infection that have not been reported convincingly prior to this manuscript. A major limitation to earlier ZIKV NHP pregnancy studies was the absence of procedure controlled uninfected animals.

This manuscript provides a detailed assessment of three different anatomic regions of the brain. The use of quantitative image analysis is a strength although the analysis could have been strengthened by evaluating increased sections from each region per animal. The small number of animals does hinder interpretation of the biological (and statistical) significance -especially since the different inoculation timepoints are grouped together which contributes to quantitative spread of the data in the image analysis.

The authors' data is supportive of a model of direct effects of viral replication on neurons in regions of the brain undergoing maturation (namely glial activation and possibly dysfunction and neuronal cell death). The rhesus macaque model is ideal to demonstrate this since regions of the macaque brain (frontal cortex) are still undergoing maturation at birth allowing for an internal control for each animal (developing versus mature brain).

There are a few stretches in interpretation of the data related to the effects of ZIKV induced immune activation and inflammation as the cause of neuronal death. Glial cells are immune cells in the brain, but there needs to be more evidence supporting the model presented that ZIKV induced inflammation and glial cell activation leads to neuronal apoptosis.

The fixation method used and sectioning method requires the use of a specialized histopathology core but the analytical methods should be able to be performed with standard image analysis tools. This paper will be useful for those doing NHP brain studies by providing a method for consistent trimming and evaluation of the brain.

The manuscript could benefit from some stylistic changes that would make the data more digestible for the non-neuropathologist.

Results:

Figure 1 and (Ln 166)-please include procedure controls for the IHC. ZIKV RNAscope for viral RNA should be included.

(Ln 213) Figure 4: 20 cells per animal, per region? This is a very small number and the findings of microstructural glial changes are very interesting-is it possible to increase the n to flesh out this data? The exact number of cells needs to be listed in the figure legend (with clarification). I think this data would benefit from showing all the cells in the graph in addition to the averages.

(Ln 234) – use of "active immune" response is inappropriate here unless there is a quantification of Iba-1 signal across groups. I do not think morphological changes can be reported as reflective of an active immune response unless this has been previously described and reported as a direct correlate to infiltrates in the brain. If so, please reference and describe better.

(Ln 237) – please indicate clearly which inoculation group had active virus in brain (all?).

Figure 7 – please include some normal for comparison here-otherwise can't interpret result without looking at supplemental figure. Would suggest including a column with just the merged from control that is in the Suppl and place next to ZIKV merged as a simple solution.

Figure 8 – see Figure 7 comments, include a high mag of normal next to high mag of Area V2 abnormal.

Figure S2 – this is exciting data and the images beautiful-can it be moved to main figures? The Iba-1 signal here is the most compelling of an "immune response" (see previous comment). Also, need to delineate the blood vessel location somehow otherwise difficult to interpret (for pathologists and non-pathologists). Maybe include H and E here? Also modify title to explain Area 46 is immature cortex. Also, please show controls at same magnification as ZIKV.

*Reviewer #3:*

Seelke, Beckman et al., describe and quantify neuroanatomical and histological abnormalities in the perinatal rhesus macaque brain following fetal infection with Zika virus (ZIKV) at three distinct gestational ages. The authors further show regional differences in pathological features in the cerebral cortex that are consistent with the caudal-to-rostral developmental timing of this structure and the known tropism of ZIKV for neural stem and progenitor cells. Specifically, macroanatomical defects are found in the occipital lobe/visual cortex, where the peak infection would have had more dramatic effect on cortical progenitor cell proliferation and survival, whereas phenotypes described in the frontal cortex are consistent with the presence of residual active infection, as nicely demonstrated by immunostaining for the virus envelope itself. The observation of combined cytoarchitectural abnormalities in both the visual thalamus (lateral geniculate nucleus, LGN) and primary visual cortex (area 17) is intriguing, suggestive of how disruptions in the retina or other more-sensitive structures/cell types might propagate up a circuit. In addition, the observations of ZIKV+ and apoptotic cells in the frontal cortex as many as 15 weeks after acute infection constitute remarkable and valuable data. Lastly, the authors provide evidence of the response of glial cell types in the brain to ZIKV, by quantitative morphological analyses of microglia and astrocytes.

Overall the authors' observations in these valuable tissues, although limited in statistical power by the small n, provide new data on the effects of ZIKV infection on brain development. However, the study is descriptive and aspects of the study, particularly with respect to affected cell types and glial cell responses, would benefit from additional cell type-specific stainings, improved imaging and presentation, and additional analysis in some places. The biggest weakness is the small sample size. Although this is a challenge with NHP studies in general, it nevertheless a weakness as reproducibility and animal to animal variability is a significant concern.

1) Additional immunofluorescent analysis of the frontal cortex in conjunction with the ZIKV staining shown in Figure 1 and caspase staining shown in Figure 4, to identify the cell types that show persistent infection many weeks after inoculation and clarify which cell types are most likely to be undergoing apoptosis at this late stage of cortical development, would further support the central argument that areal differences in pathology across the cortex are related to the caudorostral temporal developmental gradient. If the thesis is simply that ZIKV persists longer in frontal cortex because neurogenesis does as well, this could be definitively supported by such co-immunostaining.

2) Morphological changes observed in microglia of the frontal cortex are consistent with these immune cells' response to the active infection and cell death shown by the presence of ZIKV+ and caspase+ cells in this brain area. However, such morphology is quite nonspecific and the present data show correlation only, rather than suggesting any mediating, causal role for glial cell types in either the persistence or the pathological effects of Zika infection in this brain region. Without more detailed analysis of microglia state changes in the presence of Zika, it is impossible to infer whether these cells are performing their expected beneficial functions in responding to infection and tissue damage, or are additionally contributing to pathogenesis through aberrant activation or inflammatory responses.

3) The central assertion that areal differences in cortical phenotypes following Zika infection earlier in gestation are related to the temporal caudal-to-rostral gradient of cortical development is intriguing, but the manuscript would benefit from more detailed discussion and a clear model of this hypothesis, how it would arise from changes in the neural progenitor cell type composition and behaviors over cortical development, and how such interactions might be expected to result in differential structural and functional deficits in individuals infected at earlier vs later stages of fetal development.

In general this small (n=3 per group, if we ignore the different gestational ages of infection) pilot study provides additional evidence that fetal ZIKV infection disrupts brain development in ways that are influenced by the normal gradient of regional and cortical areal neurogenesis -- e.g. caudal-to-rostral in the cortex. Combined cytoarchitectural disruptions of the LGN, V1, and occipital gyrification are intriguing, suggesting that disruptions in the retina or other more-sensitive structures/cell types might propagate up a circuit, although no mechanistic interpretation is provided and further discussion of how such propagation might occur would be nice. In addition, the observation of ZIKV+ and apoptotic cells in the frontal cortex so long after the acute infection is a remarkable and valuable piece of data. The study would be greatly strengthened by additional immunohistochemical analyses of the frontal cortex to determine the cellular identity of ZIKV+ and apoptotic cells, and/or the spatial distribution of ZIKV+ cells relative to apoptotic neurons or other cell types. The authors' assertion that their methods reveal "where the virus was found, [and] what cell types were affected" is not fully supported by the data provided, given that ZIKV+ cells are not co-stained for any markers and their location/distribution is not described beyond the fact that they are present in both grey and white matter.

The glial angle of the work is lacking in strong phenotypes or clear mechanistic insight. Microglia morphological changes in the frontal cortex are a clear but non-specific observation, and not terribly surprising in proximity to active virus and/or apoptosis (although how proximal, again, is not clear without additional images or co-stainings). Although a relationship of the microglial phenotype to the increased cell death is implied, no mechanistic hypothesis is explicitly stated, and it is just as likely that microglia would become activated in response to cell death rather than cause it. Macroscopic phenotypes as observed in occipital cortex and LGN are even further removed from any potential glial mechanisms and in fact are quite similar to phenotypes described to arise from various non-viral/genetic/non-inflammatory disruptions of NPC survival, proliferation, and differentiation.

In general, the authors suggest or imply mechanisms which could stand to be more explicitly stated as conjectures or hypotheses supported by the current data and/or subject to future study. In addition to the specific role of microglia, the impact of the temporal caudal-rostral developmental gradient on the final neuropathology following infection at distinct gestational ages is not clearly explained or discussed. One gets the sense that reduced gyrification in occipital but not frontal cortex, or absence of changes to cortical thickness, could be related to the earlier development of occipital cortex, but exactly how this would occur -- e.g. through differential effects on subtypes of neural stem/progenitor cells and their relationships to tangential vs radial expansion of cortical plate. An explicit model of how regional phenotypic differences in cortex could arise given infection at different fetal stages would be a nice addition to the manuscript. (It's not clear why gyrification should not also be reduced in frontal lobe if infection was present at the earlier gestational ages when ventricular progenitors are highly proliferative -- possibly because the frontal lobe has not completed folding at birth in macaques?)

Even with further and clearer discussion of interpretations and model, I do not believe the paper merits publication in *eLife* without additional experiments, primarily co-immunostaining of ZIKV and CC3 with each other and with NPC and neuronal cell type markers.

Specific comments and suggestions to strengthen study (using existing tissue/samples).

Figure 1. Main finding: ZIKV is found in frontal but not occipital ctx nor in subcortical areas at near-term, after inoculation weeks earlier.

– What are the infected cell types? *Sox2*/NeuN/ZIKV co-stain would go far in supporting the argument that different phenotypes observed between frontal and occipital lobes relates to the late development of frontal lobe and persistence of NPC in this area, allowing ZIKV to continue to proliferate.

– A lower magnification view, or some quantifications, illustrating the frequency/density/laminar distribution of ZIKV+ cells in the frontal lobe would be very helpful in inferring the potential downstream impacts of persistent infection in this brain region.

– The term "activated" in the Figure 1 legend and associated Results section to describe astrocytes and microglia should be defined here, or simply removed given later discussion at Figure 4. In fact, text and figure legend here refer to "activated astrocytes" and "reactive astrocytes," but the analysis shown in Figure 4 revealed no significant differences in frontal lobe astrocytes between ZIKV+ and control, a discrepancy that should be addressed.

Figure 2. Main finding: ZIKV reduced gyrification in occipital but not frontal lobe, and caused no significant changes in total, GM, or WM volumes.

–To my knowledge a more commonly reported (though not necessarily better) local gyrification measurement is the ratio of outer cortical surface area to the "convex hull" area.

- the curve that encloses the exposed cortex, excluding sulci. See for example https://pubmed.ncbi.nlm.nih.gov/20176115/

Since this is a more commonly reported GI and quite straightforward/quick to implement with the existing images it would be nice to report this measurement in addition to what's already included.

– Given the abstract and introduction's emphasis on the rostrocaudal developmental timing of the cortex, the authors should further discuss why they think such timing might result in reduced gyrification in occipital lobe, but not frontal.

Figure 3. Main finding: no changes in cortical thickness. Again, there could perhaps be some discussion of the motivation for taking this measurement, or placing this finding in the context of the larger argument about developmental timing.

Figure 4. Main finding: frontal lobe microglia in ZIKV+ brains show smaller whole cell volume and fewer terminal branch points, suggestive of an activated state.

– Following on the comment to Figure 1 above, the main text and figure 4 legend need to be aligned - e.g. the figure legend states "astrocytes tended to have smaller cell body volumes (R)" but the main text reports this comparison as having a p-value of 0.37 and states astrocytes "did not differ between ZIKV-infected and control animals."

– If there is no statistically significant difference in astrocyte morphology, density, or distribution in ZIKV+ cortex, the terms "activated" or "reactive" astrocytes should be removed.

Figure 5. Main finding: increased number of caspase-positive neurons in cortical plate of area 46 of ZIKV+ brains.

– The following sentence/section should be rewritten to clarify the hypothesis being tested: "IHC fluorescent microscopy analyses for cleaved caspase-3 and SATB2, which are markers of apoptosis onset and immature cortical neurons, respectively, were carried out in order to determine whether NPCs underwent higher rates of apoptosis in immature neurons induced by the presence of ZIKV in the brain in both Area 17 and Area 46." Satb2 is a neuronal marker whereas previously apoptosis was observed in NPCs (progenitors, e.g. *Sox2*^+^ cells). ZIKV was present in area 46 but not 17 at the time of analysis.

– Are any of the caspase-positive SATB2+ neurons also ZIKV+? Are there any SATB2-negative caspase+ cells? Are there more total CC3+ cells in ZIKV+ vs control? Please be more specific in the units of measurement shown on the y-axis -- number of cells per what area/volume? It would also be helpful to report these data as total density of caspase+ cells regardless of SATB2 immunoreactivity, as well as the proportions of SATB2+ cells that are caspase+, and vice versa. Is there a preferential increase in apoptosis of neurons, or a general increase in apoptosis affecting multiple cell types?

– *Sox2*^+^ glial progenitor cells derived from earlier VZ/SVZ NPC are abundant across the cortical plate at late gestation. Given the previous reports of NPC infection by ZIKV, did the authors also observe caspase+ *Sox2*^+^ cells? (Again, what is the identity of the ZIKV+ cells shown in Figure 1?)

– If the apoptotic neurons are not themselves ZIKV+, what is the proposed mechanism/cause of their apoptosis?

Figure 6. Main finding: blurred laminar boundaries/absence of koniocellular layers in ZIKV+ LGN.

– Please include an un-pseudocolored version of the control section to allow the reader to better see what the normal layering pattern should look like across the whole LGN.

Figure 7. Main finding: reduced neuronal density, increased "glia activation" in dysmorphic areas of LGN.

– The phenotype asserted in the main text, "the density of the neuronal population (stained with NeuN) in these areas was also substantially reduced," is not immediately obvious from the images. It would help to indicate more precisely on the Nissl image the area shown at higher mag in IFL panels. This should be roughly achievable by comparing the Nissl and DAPI on the adjacent sections. Better yet would be to show a box on panels G-K to indicate the areas shown in B-F and L-P.

– Most convincing would be some quantification of NeuN+ cell density or cumulative signal in the dysmorphic ROI's, compared either to neighboring unaffected regions of LGN or to matched ROI's from the control LGNs.

– The assertion of "glia activation" is far from obvious given only the images in Figure 7. Quantitative morphological analysis of microglia similar to what was performed for cortex (Figure 4) would be a big help. Just improving the clarity of the images in panels C, D, M, and N would also be helpful; it is difficult to make out for example the increase in GFAP content in the NeuN-poor region of panel N.

---

## [Author Response]

Reviewer #1:Children born during the Zika outbreak are known to have marked changes in neurological function and in some cases exhibit microcephaly, but the anatomical bases for these deficits is unknown. The goal of this study was to identify neuropathologies in near-term monkeys whose mothers were infected with Zika (ZIKV) during gestation. Using a combination of histological and immunofluorescent techniques, the authors demonstrate that compared to control monkeys, the ZIKV-infected fetuses exhibit a number of anatomically discrete abnormalities that are accompanied by evidence of localized neuro-immune responses and cell death. These abnormalities are shown through a comprehensive series of fluorescent micrographs that make the differences between the infected and control monkeys clear to the reader. The quantification methodologies are explained in detail, and the authors take care to note statistical considerations for the small sample size. Importantly, the observations map onto areas whose functions are affected in reports of children infected with ZIKV in utero, and therefore this work could have a major impact on public health.No recommendations, this is a very nicely-prepared paper that needs no improvements.

Thank you so much for the positive feedback about our manuscript.

Reviewer #2:[…]The manuscript could benefit from some stylistic changes that would make the data more digestible for the non-neuropathologist.Results:Figure 1 and (Ln 166)-please include procedure controls for the IHC.

We thank the reviewer for the commentaries and suggestions to improve our manuscript. To better demonstrate that ZIKV induced glia abnormalities in Area 46 in comparison with control monkeys, we completely modified Figure 1. The first panel of Figure 1 shows now representative micrographs of the same region in Area 46 from a control and a ZIKV-infected animal. A second and a third panel were also added in figure 1, highlighting that Iba1+ microglia involved in the neuroinflammatory abnormalities observed in the ZIKV-infected monkeys, also expresses CD68, a lysosomal protein highly expressed in activated microglia. A new section titled: ‘Neuroinflammatory abnormalities and ZIKV presence in brains from ZIKV-infected animals”, was also added in the “Results” section, starting on Line 161.

It reads:

“Immunohistochemical (IHC) confocal microscopy revealed the presence of clusters of reactive glia within Area 46 of the PFC from ZIKV+ animals, but not in the controls (Figure 1A-B). Multiple labelling fluorescent microscopy allowed us to visualize that abnormal clusters of Iba1+ microglia which also express the lysosomal marker CD68 in the ZIKV-infected animals (Figure 1C-G). CD68 levels are known to be substantially upregulated in microglia during inflammatory processes and are thought to be involved in active phagocytosis (46). We also observed that clusters of amoeboid-shaped microglia within Area 46, are directly engulfing neuronal debris in the ZIKV-infected monkeys (Figure 1H-I, 3D reconstruction: J, K).

Next, we sought to detect the presence of ZIKV protein in the same regions we observed active inflammatory response. Interestingly, we observed the presence of ZIKV envelope protein within Area 46 of all the infected animals, with detection of viral protein inside clusters of microglia, but not astrocytes (GFAP, Figure 2A-G, 3D reconstruction: 2H). Despite the fact that cells expressing ZIKV were detected in Area 46, we were not able to confirm the identity of the infected cells. As shown in Figure 2I-J, these ZIKV+ cells are thought to be in a direct process of phagocytosis, a common event occurring during viral encephalitis (47). While we could not detect the expression of other proteins in the ZIKV+ cells, they were surrounded and in direct contact with reactive microglia also expressing CD68, as shown in Figure 2K-L.”

ZIKV RNAscope for viral RNA should be included.

Unfortunately, due to the protocol used for tissue fixation and preparation, we were unable to detect viable RNA for performing ZIKV RNAscope investigation. While there are groups who have used RNAscope in fixed tissue, many groups have been unsuccessful in processing fixed tissue in this way. Given this, we are electing to use evidence from immunohistochemistry demonstrating presence of Zika antibodies.

(Ln 213) Figure 4: 20 cells per animal, per region? This is a very small number and the findings of microstructural glial changes are very interesting-is it possible to increase the n to flesh out this data? The exact number of cells needs to be listed in the figure legend (with clarification). I think this data would benefit from showing all the cells in the graph in addition to the averages.

All the images for glia morphological analysis were acquired and processed using 3D high-resolution volumetric analysis. The number of 20 cells per animal per region was described recently by our group to be enough to detect subtle changes in glia morphology induced by inflammatory events (Beckman et al., PNAS 2019, http://doi.org/10.1073/pnas.1902301116, Beckman et al., Alzheimer’s and Dementia, 2021, https://doi.org/10.1002/alz.12318). Due to the fact that the mentioned studies were performed using middle-aged rhesus monkey brain, instead of the perinatal brains analyzed in the current study, we increased the number of glial cells analyzed to 28 cells per animal, per region. The exact number of cells analyzed and details about image processing were added in the legend of the figure. While we appreciate that the reviewer would prefer to see all of the cells this would create a huge figure (28 x 2 (astrocytes, microglia) X 2 brain areas (area 17 and 46) X 6 animals) and we think that the main story communicated by the example cells and statistical figure would not be clear from that figure. The graphs from all other figures only present a summary of the data collected from each animal, instead of all cells or micrographs analyzed. That way, we believe keeping the graph with this same visual identity in figure 4 will facilitate for the reader, comparisons between groups and figures. In addition, we included, as suggested by reviewer 3, a new image, more representative, for reactive microglia within area 46 of the ZIKV infected animals.

(Ln 234) – use of "active immune" response is inappropriate here unless there is a quantification of Iba-1 signal across groups. I do not think morphological changes can be reported as reflective of an active immune response unless this has been previously described and reported as a direct correlate to infiltrates in the brain. If so, please reference and describe better.

We agree with the reviewer that the term “active immune response” might induce the establishment of a connection between our observation and direct findings of infiltrates in the brain, which we cannot conclude based on the current data. We modified the beginning of sentence for “A neuroinflammatory response, mainly driven by CD68/Iba+ microglia” (Line 252). As mentioned in the previous reply, we and others have shown that alterations in microglia morphology can indicate a direct response of these cells to inflammatory events in the brain. We believe that by combining now the general marker Iba-1 with the marker for activation CD68, we were able to show more clearly the direct involvement of activated microglia in phagocytizing ZIKV protein in the frontal lobe.

That section that begins on Line 252 now reads:

“Microstructural changes related to neural development. A neuroinflammatory response, mainly driven by CD68/Iba+ microglia, in combination with the presence of ZIKV envelope protein, indicated that the frontal lobe for all ZIKV-infected subjects was a site of persistent ZIKV infection for a minimum of 60 days post-inoculation. […] The higher frequency of immature neuron death, the active immune response (activated microglia), and the presence of ZIKV protein in Area 46 suggest that frontal cortex remained a site of active ZIKV-induced neuronal remodeling at the time the brains were analyzed.”

(Ln 237) – please indicate clearly which inoculation group had active virus in brain (all?).

All three Zika infected animals analyzed in the study had evidence of active virus in their brains. We have now indicated that on Line 172 and 313 of the manuscript.

Figure 7 – please include some normal for comparison here-otherwise can't interpret result without looking at supplemental figure. Would suggest including a column with just the merged from control that is in the Suppl and place next to ZIKV merged as a simple solution.

We agree with the reviewer that showing side-by-side normal vs. zika infected images is important for understanding and interpreting the results. In the mentioned figure (previously Figure 7, now Figure 8), we have side-by-side representative images of Nissl and confocal images of LGN from both CTR and ZIKV groups.

Figure 8 – see Figure 7 comments, include a high mag of normal next to high mag of Area V2 abnormal.

High magnification representative images of normal V1/V2 transition vs. layer discontinuity and V2 abnormality, are now included side-by-side in the new figure (current Figure 9).

Figure S2 – this is exciting data and the images beautiful-can it be moved to main figures? The Iba-1 signal here is the most compelling of an "immune response" (see previous comment). Also, need to delineate the blood vessel location somehow otherwise difficult to interpret (for pathologists and non-pathologists). Maybe include H and E here? Also modify title to explain Area 46 is immature cortex. Also, please show controls at same magnification as ZIKV.

We thank the reviewer for the suggestion of moving Figure S2 (ZIKV neuroinflammatory response in area 46) to the main manuscript. As mentioned in the reply to comment 1A, we completely modified main figures 1 and 2 in the manuscript to highlight the details of how ZIKV infection is inducing a neuroinflammatory response driven by activated microglia (Iba+/CD68+), that is not present in the control animals. The title explaining Area 46 is immature cortex was also added in the legend of Figure 1.

Reviewer #3:[…]In general this small (n=3 per group, if we ignore the different gestational ages of infection) pilot study provides additional evidence that fetal ZIKV infection disrupts brain development in ways that are influenced by the normal gradient of regional and cortical areal neurogenesis -- e.g. caudal-to-rostral in the cortex. Combined cytoarchitectural disruptions of the LGN, V1, and occipital gyrification are intriguing, suggesting that disruptions in the retina or other more-sensitive structures/cell types might propagate up a circuit, although no mechanistic interpretation is provided and further discussion of how such propagation might occur would be nice. In addition, the observation of ZIKV+ and apoptotic cells in the frontal cortex so long after the acute infection is a remarkable and valuable piece of data. The study would be greatly strengthened by additional immunohistochemical analyses of the frontal cortex to determine the cellular identity of ZIKV+ and apoptotic cells, and/or the spatial distribution of ZIKV+ cells relative to apoptotic neurons or other cell types. The authors' assertion that their methods reveal "where the virus was found, [and] what cell types were affected" is not fully supported by the data provided, given that ZIKV+ cells are not co-stained for any markers and their location/distribution is not described beyond the fact that they are present in both grey and white matter.

We appreciate the comment from the reviewer and understand how useful it would be to know what cell types are particularly subject to apoptosis during fetal Zika virus infection. We did carry out some of these analyses, attempting to co-localize markers of apoptosis with other markers – for example our CC3 and SATB2 co-localization analyses give us information about the death of immature neurons. Unfortunately, in many cases, once apoptosis begins and certainly once it is underway, identification of the cells using IHC (via surface markers) does not paint a comprehensive picture of the system because the cells literally fall apart and so we were limited in terms of what approaches were successful. We agree that this is a potentially fruitful avenue for future research where tissue is processed at a different time point of infection (i.e., our time point is ~100 days post infection).

The glial angle of the work is lacking in strong phenotypes or clear mechanistic insight. Microglia morphological changes in the frontal cortex are a clear but non-specific observation, and not terribly surprising in proximity to active virus and/or apoptosis (although how proximal, again, is not clear without additional images or co-stainings). Although a relationship of the microglial phenotype to the increased cell death is implied, no mechanistic hypothesis is explicitly stated, and it is just as likely that microglia would become activated in response to cell death rather than cause it. Macroscopic phenotypes as observed in occipital cortex and LGN are even further removed from any potential glial mechanisms and in fact are quite similar to phenotypes described to arise from various non-viral/genetic/non-inflammatory disruptions of NPC survival, proliferation, and differentiation.

It is the case that our data do not allow us to determine whether inflammatory processes are causing or are caused by cell death, and we have now indicated this in the Discussion (Lines 398-400). It reads:

“The histological analyses presented here differ in that we measured specific cytoarchitectonic regions of the brain as well as morphological analyses of glial cells in the frontal and occipital cortex. Together, these analyses paint a picture of inflammatory processes related to viral infiltration sweeping through the cortex and a wave of cell death. Given our analyses, it is not possible to tell whether inflammatory processes are caused by or are causing cell death, but they certainly appear to co-occur given their proximity.”

We have also added a discussion of the relationship between morphology and activation in the Results section, beginning at Line 216. It reads:

“Microstructural Changes to Glia. Following macrostructural evaluations, we carried out a series of immunohistochemical analyses to quantify cell-level features that might be impacted by ZIKV infection with a specific focus on glia (microglia and astrocytes) in Brodmann’s Area 17 and Area 46. […] Iba1 and GFAP were used as general microglia and astrocytes markers, respectively, and each cell was exported and analyzed individually for total and cell body volumes, and for number of terminal point (branching ramification).”

In general, the authors suggest or imply mechanisms which could stand to be more explicitly stated as conjectures or hypotheses supported by the current data and/or subject to future study. In addition to the specific role of microglia, the impact of the temporal caudal-rostral developmental gradient on the final neuropathology following infection at distinct gestational ages is not clearly explained or discussed. One gets the sense that reduced gyrification in occipital but not frontal cortex, or absence of changes to cortical thickness, could be related to the earlier development of occipital cortex, but exactly how this would occur -- e.g. through differential effects on subtypes of neural stem/progenitor cells and their relationships to tangential vs radial expansion of cortical plate. An explicit model of how regional phenotypic differences in cortex could arise given infection at different fetal stages would be a nice addition to the manuscript. (It's not clear why gyrification should not also be reduced in frontal lobe if infection was present at the earlier gestational ages when ventricular progenitors are highly proliferative -- possibly because the frontal lobe has not completed folding at birth in macaques?)Even with further and clearer discussion of interpretations and model, I do not believe the paper merits publication in eLife without additional experiments, primarily co-immunostaining of ZIKV and CC3 with each other and with NPC and neuronal cell type markers.

We appreciate the comments from the reviewer, and in order to address these questions we ran a new set of IHC experiments with multi-labeling fluorescence combinations. A completely modified Figure 1 and a whole new Figure 2 were added. The new micrographs included in Figure 1 highlight: 1 – The general glia alterations observed in the prefrontal cortex of ZIKV infected animals but not in control animals (Figure 1A-B); 2 – Inflammatory clusters observed in the ZIKV tissue were formed by reactive astrocytes and microglia, with the latest also co-expressing CD68, a lysosomal protein expressed in phagocytic cells (Figure 1C-G); 3 – 3D analysis of the reactive microglia from the ZIKV animals show these cells are actively engulfing apoptotic material formed by fragmented Dapi and NeuN (Figure 1H-K).

To address the identity of the ZIKV+ cells, we ran several multi-label antibodies combinations targeting glia, neurons, and apoptotic cells. Figure 2 panels now show that reactive microglia and astrocytes are surrounding ZIKV+ cells, and that ZIKV protein is observed inside Iba+/CD68+ microglia. Importantly, panel I-J shows in detail the 3D reconstruction of how microglia is interacting with ZIKV+ cells. After several rounds of IHC with different multi-labeling combinations, we cannot clearly point to an identity for the main ZIKV+ cells within these regions. We believe these cells are in an advanced cell death/phagocytosis stage, when protein synthesis is impaired inside the cell, and as consequence, labeling them becomes a real challenge, especially with limitations in the tissue fixation after necropsy. In addition, we do not observe a clear pattern of location/distribution of these infected cells within Area 46, besides the fact that they are observed in both, white and gray matter. Clear imaging of these cells can only be done using high-resolution microscopy, as often these cells are not expressing other markers, including DAPI. Imaging the entire Area 46 of the monkeys used in this study using the 63 or 100x objective to quantify the distribution and location of these cells would generate hundreds of images and an extensive amount of time to analyze it, that is beyond the scope of this project. But we expect that the new data added in the figures provides more evidence of the persistent of the virus within specific areas of the brain, and how this can reflect in neuroinflammatory abnormalities and neuronal loss, even after longer periods of viral incubation.

Specific comments and suggestions to strengthen study (using existing tissue/samples).Figure 1. Main finding: ZIKV is found in frontal but not occipital ctx nor in subcortical areas at near-term, after inoculation weeks earlier.– What are the infected cell types? Sox2/NeuN/ZIKV co-stain would go far in supporting the argument that different phenotypes observed between frontal and occipital lobes relates to the late development of frontal lobe and persistence of NPC in this area, allowing ZIKV to continue to proliferate.– A lower magnification view, or some quantifications, illustrating the frequency/density/laminar distribution of ZIKV+ cells in the frontal lobe would be very helpful in inferring the potential downstream impacts of persistent infection in this brain region.– The term "activated" in the Figure 1 legend and associated Results section to describe astrocytes and microglia should be defined here, or simply removed given later discussion at Figure 4. In fact, text and figure legend here refer to "activated astrocytes" and "reactive astrocytes," but the analysis shown in Figure 4 revealed no significant differences in frontal lobe astrocytes between ZIKV+ and control, a discrepancy that should be addressed.

Related to the cell type of the ZIKV infected cells, as we mentioned in the previous reply, we ran several IHC experiments with different antibodies combination, including *Sox2*/NeuN/ZIKV, as suggested by the reviewer. Fragments of both neuronal markers were found inside phagocytic microglia in direct contact with ZIKV+ cells, but not inside the infected cells the ZIKV+ cell (Figures 1 and 2).

We agree with the reviewer that understanding the frequency/density/laminar distribution of ZIKV+ cells in the frontal lobe would provide deeper understanding of the viral impact in the normal cortical development. Our intention is to do a detailed stereological quantification of the Zika viral distribution in the frontal lobe of the rhesus monkeys, but that is beyond the scope of this paper, this paper is intended to be an initial description of the pathology observed in a recent developed monkey model. We believe that for this type of analysis to be done correctly, substantial amount of time and work will be required, especially due to the size and complexity of this cerebral region in the monkey brain.

In relation to the use of the term “activated” in Figure 1 legend, we agree that there is no indication of astrocyte activation, and the term should not be used in relation to the astrocyte’s morphology. All mentions to the term “activated” in relation to astrocytes were removed from the text.

Figure 2. Main finding: ZIKV reduced gyrification in occipital but not frontal lobe, and caused no significant changes in total, GM, or WM volumes.–To my knowledge a more commonly reported (though not necessarily better) local gyrification measurement is the ratio of outer cortical surface area to the "convex hull" area.- the curve that encloses the exposed cortex, excluding sulci. See for example https://pubmed.ncbi.nlm.nih.gov/20176115/Since this is a more commonly reported GI and quite straightforward/quick to implement with the existing images it would be nice to report this measurement in addition to what's already included.– Given the abstract and introduction's emphasis on the rostrocaudal developmental timing of the cortex, the authors should further discuss why they think such timing might result in reduced gyrification in occipital lobe, but not frontal.

We thank the reviewer for the suggestion. We analyzed the local gyrification index in the frontal and occipital lobes using the method described by Rogers at al 2010 (the paper mentioned by the reviewer). We analyzed between 14 and 19 sections for each animal in each group but found no difference in the cerebral gyrification index between the groups for both regions of the cortex. As such, we’ve maintained our original presentation of the gyrification data.

Figure 3. Main finding: no changes in cortical thickness. Again, there could perhaps be some discussion of the motivation for taking this measurement, or placing this finding in the context of the larger argument about developmental timing.

Changes of cortical thickness during development might reflect spines/dendrites loss and ultimately neuronal death. In microencephaly induced by ZIKV infection, the loss of cortical white and gray matter can be so substantial that cortical thickness measurement can be used to indicative the degree of the pathology. The investigation of cortical thickness in the monkeys using Nissl was performed in combination with the fluorescent microscopy experiments investigating increased cell death of immature neurons in the frontal lobe of ZIKV animals. We believe is important to show that despite the fact that there is no microencephaly cases between the infected monkeys, nor a change in the cortical thickness, there are significant micro-structural changes such as increased death of immature neurons and robust neuroinflammatory response, that ultimately can contribute to long-term, chronic impairment related to early-development ZIKV infection.

Figure 4. Main finding: frontal lobe microglia in ZIKV+ brains show smaller whole cell volume and fewer terminal branch points, suggestive of an activated state.– Following on the comment to Figure 1 above, the main text and figure 4 legend need to be aligned - e.g. the figure legend states "astrocytes tended to have smaller cell body volumes (R)" but the main text reports this comparison as having a p-value of 0.37 and states astrocytes "did not differ between ZIKV-infected and control animals."– If there is no statistically significant difference in astrocyte morphology, density, or distribution in ZIKV+ cortex, the terms "activated" or "reactive" astrocytes should be removed.

We agree with the reviewer, and as mentioned before, all mentions to “activation” in relation to astrocytes, were removed from the main text and figure legends.

Figure 5. Main finding: increased number of caspase-positive neurons in cortical plate of area 46 of ZIKV+ brains.– The following sentence/section should be rewritten to clarify the hypothesis being tested: "IHC fluorescent microscopy analyses for cleaved caspase-3 and SATB2, which are markers of apoptosis onset and immature cortical neurons, respectively, were carried out in order to determine whether NPCs underwent higher rates of apoptosis in immature neurons induced by the presence of ZIKV in the brain in both Area 17 and Area 46." Satb2 is a neuronal marker whereas previously apoptosis was observed in NPCs (progenitors, e.g. Sox2^+^ cells). ZIKV was present in area 46 but not 17 at the time of analysis.– Are any of the caspase-positive SATB2+ neurons also ZIKV+? Are there any SATB2-negative caspase+ cells? Are there more total CC3+ cells in ZIKV+ vs control? Please be more specific in the units of measurement shown on the y-axis -- number of cells per what area/volume? It would also be helpful to report these data as total density of caspase+ cells regardless of SATB2 immunoreactivity, as well as the proportions of SATB2+ cells that are caspase+, and vice versa. Is there a preferential increase in apoptosis of neurons, or a general increase in apoptosis affecting multiple cell types?– Sox2^+^ glial progenitor cells derived from earlier VZ/SVZ NPC are abundant across the cortical plate at late gestation. Given the previous reports of NPC infection by ZIKV, did the authors also observe caspase+ Sox2^+^ cells? (Again, what is the identity of the ZIKV+ cells shown in Figure 1?)– If the apoptotic neurons are not themselves ZIKV+, what is the proposed mechanism/cause of their apoptosis?

We agree with the reviewer, and the sentence mentioned was rewritten to address that IHC fluorescent microscopy for cleaved caspase 3 was conducted in order to investigate if the increase in reactive microglia population and active presence of ZIKV can induce an increase in the number of immature neurons (SATB2) undergoing apoptosis (Ln 271-276). We carried out additional IHC combinations in order address the reviewer questions, and an updated version of the figure mentioned (currently Figure 6) and new Supplementary Figure 3 were added to the manuscript. The updated Figure 6 includes now the total cell counting for each of the markers analyzed, DAPI (total number of cells), SATB2 (total number of immature neurons) and CC3 (total number of apoptotic cells). A separate graph analyzing SATB2/CC3 colocalization was also added in the figure, for each region analyzed. Despite the fact that there is no significant difference in the total number of CC3 cells across the two analyzed regions between control and ZIKV animals, an increase in SATB2/CC3 was observed in the ZIKV animals in Area 46, but not 17. A deeper investigation of these regions with higher-resolution image acquisition showed that activated microglia (HLA-Dr+ cells) also express CC3 in the region, especially the cells in direct contact with neurons, possibly during a phagocytosis process. Images for this analysis are show in the new supplementary figure 3. Finally, once again we carry out different experiments to address if ZIKV+ is present in these SATB2/CC3 or HLA-Dr/CC3 cells, but once again, the virus could not be detected in these cells. Whether this is caused by loss of viral epitopes during the brain fixation process, or due to the stage of the viral infection/engulfment by microglia per se, could not be address at the time for this manuscript.

Figure 6. Main finding: blurred laminar boundaries/absence of koniocellular layers in ZIKV+ LGN.– Please include an un-pseudocolored version of the control section to allow the reader to better see what the normal layering pattern should look like across the whole LGN.Figure 7. Main finding: reduced neuronal density, increased "glia activation" in dysmorphic areas of LGN.

We thank the reviewer for the suggestion. To better allow the reader to compare between normal and altered layers of the LGN, we removed the coloring patterns from the micrographs. We concluded that the coloring pattern applied before was not necessarily and removed from the figure.

– The phenotype asserted in the main text, "the density of the neuronal population (stained with NeuN) in these areas was also substantially reduced," is not immediately obvious from the images. It would help to indicate more precisely on the Nissl image the area shown at higher mag in IFL panels. This should be roughly achievable by comparing the Nissl and DAPI on the adjacent sections. Better yet would be to show a box on panels G-K to indicate the areas shown in B-F and L-P.– Most convincing would be some quantification of NeuN+ cell density or cumulative signal in the dysmorphic ROI's, compared either to neighboring unaffected regions of LGN or to matched ROI's from the control LGNs.– The assertion of "glia activation" is far from obvious given only the images in Figure 7. Quantitative morphological analysis of microglia similar to what was performed for cortex (Figure 4) would be a big help. Just improving the clarity of the images in panels C, D, M, and N would also be helpful; it is difficult to make out for example the increase in GFAP content in the NeuN-poor region of panel N.

We agree with the reviewer, and the mentioned Figure (currently Figure 8) was redone to better show how the Nissl images correlate with the confocal images between controls and ZIKV animals. A “zoom” area was also added for both groups to highlight the differences observed between the groups. We agree with the reviewer that quantification of NeuN and Iba1 count would be informative, but require a substantial amount of time, and are beyond the scope of this manuscript now. Finally, we also agree that the assertion “glia activation” is not obvious, so besides improving the clarity of the images in the figure, we also removed the term “glia activation” from the figure legend and related text in the Result sections.